# Myogenin controls via AKAP6 non-centrosomal microtubule-organizing center formation at the nuclear envelope

**Robert Becker[1], Silvia Vergarajauregui[1], Florian Billing[1], Maria Sharkova[1], Eleonora Lippolis[2], Kamel Mamchaoui[3], Fulvia Ferrazzi[2,4,5], Felix B Engel[1,6]***

[1]Experimental Renal and Cardiovascular Research, Department of Nephropathology, Institute of Pathology, Friedrich-Alexander-Universität Erlangen-Nürnberg (FAU), Erlangen, Germany; [2]Institute of Human Genetics, Friedrich-Alexander-Universität Erlangen-Nürnberg, Erlangen, Germany; [3]Sorbonne Universités UPMC Université Paris 06, INSERM U974, CNRS FRE3617, Center for Research in Myology, GH Pitié Salpêtrière, 47 Boulevard de l'Hôpital, Paris, France; [4]Department of Nephropathology, Institute of Pathology, Friedrich-Alexander-Universität Erlangen-Nürnberg (FAU), Erlangen, Germany; [5]Institute of Pathology, Friedrich-Alexander-Universität Erlangen-Nürnberg (FAU), Erlangen, Germany; [6]Muscle Research Center Erlangen (MURCE), Erlangen, Germany

**\*For correspondence:**
felix.engel@uk-erlangen.de

**Competing interest:** The authors declare that no competing interests exist.

**Abstract** Non-centrosomal microtubule-organizing centers (MTOCs) are pivotal for the function of multiple cell types, but the processes initiating their formation are unknown. Here, we find that the transcription factor myogenin is required in murine myoblasts for the localization of MTOC proteins to the nuclear envelope. Moreover, myogenin is sufficient in fibroblasts for nuclear envelope MTOC (NE-MTOC) formation and centrosome attenuation. Bioinformatics combined with loss- and gain-of-function experiments identified induction of AKAP6 expression as one central mechanism for myogenin-mediated NE-MTOC formation. Promoter studies indicate that myogenin preferentially induces the transcription of muscle- and NE-MTOC-specific isoforms of *Akap6* and *Syne1*, which encodes nesprin-1α, the NE-MTOC anchor protein in muscle cells. Overexpression of AKAP6β and nesprin-1α was sufficient to recruit endogenous MTOC proteins to the nuclear envelope of myoblasts in the absence of myogenin. Taken together, our results illuminate how mammals transcriptionally control the switch from a centrosomal MTOC to an NE-MTOC and identify AKAP6 as a novel NE-MTOC component in muscle cells.

## Introduction

Correct organization of the microtubule cytoskeleton is essential for many cellular processes such as the establishment of cell shape, organelle positioning, or intracellular transport (*Akhmanova and Steinmetz, 2015*; *Conduit et al., 2015*). In proliferating vertebrate cells, proteins that control microtubule nucleation and anchoring accumulate as pericentriolar material (PCM) at the centrosome, which in turn functions as the dominant microtubule-organizing center (MTOC) (*Prosser and Pelletier, 2015*). The centrosomal MTOC is pivotal for cell cycle progression and correct chromosome segregation during mitosis (*Hinchcliffe et al., 2001*; *Khodjakov and Rieder, 2001*; *Sir et al., 2013*). In contrast, MTOC function is assigned to non-centrosomal sites (ncMTOCs) during differentiation of various cell types (*Sanchez and Feldman, 2017*). In epithelial cells, apically localized ncMTOCs participate in organelle positioning and help to establish apical-basal cell polarity (*Brodu et al., 2010*; *Lee et al., 2007*; *Meads and Schroer, 1995*; *Toya et al., 2016*). In neurons, dendritic branch points, Golgi

outposts, and preexisting microtubules have been suggested to act as ncMTOC sites and precise control of microtubule array polarity helps to define the axonal and dendritic compartments (*Luders, 2020*; *Nguyen et al., 2014*; *Ori-McKenney et al., 2012*; *Sanchez-Huertas et al., 2016*). In striated (i.e., heart and skeletal) muscle cells, ncMTOCs form at the nuclear envelope and contribute to correct nuclei positioning in skeletal myotubes (*Elhanany-Tamir et al., 2012*; *Espigat-Georger et al., 2016*; *Gimpel et al., 2017*; *Kronebusch and Singer, 1987*; *Tassin et al., 1985*). Notably, human myopathies, such as centronuclear myopathy (CNM) or Emery–Dreifuss muscular dystrophy (EDMD), often feature mislocalized nuclei (*Jungbluth and Gautel, 2014*; *Madej-Pilarczyk and Kochański, 2016*). While the nuclear envelope MTOC (NE-MTOC) has been implicated in early steps of myonuclear positioning in vitro, a direct link between NE-MTOC defects and human myopathies has not been established, possibly due to the fact that many aspects of NE-MTOC formation and function remain unclear. Similarly, while microtubules are important regulators of contractility and nuclear architecture in cardiomyocytes (*Chen et al., 2018*; *Heffler et al., 2020*), the specific role of NE-MTOC-generated microtubules remains unclear.

Despite progress in illuminating identity, composition, and function of ncMTOCs, it remains elusive which mechanisms initiate the switch from centrosomal to non-centrosomal MTOCs during differentiation of vertebrate cells. The only mechanistic insight into ncMTOC induction has been gained by studying *Drosophila* tracheal cells. It was shown that the transcription factor trachealess, which specifies tracheal fate, is required for the spastin-mediated release of centrosomal components from the centrosome and their subsequent Piopio-mediated anchoring to the apical membrane (*Brodu et al., 2010*).

Here, we aimed to identify mechanisms that control NE-MTOC formation in mammals utilizing skeletal muscle differentiation as an experimental system. Mammalian skeletal muscle differentiation is controlled by a family of transcription factors termed myogenic regulatory factors (MRFs) (*Braun and Gautel, 2011*; *Buckingham and Rigby, 2014*). Among those, myoblast determination protein (MyoD) regulates commitment to a myogenic fate and is thought to promote early differentiation of myoblasts (*Comai and Tajbakhsh, 2014*; *Ishibashi et al., 2005*). The MRF myogenin acts as a unique regulator of terminal differentiation during myogenesis. In the absence of myogenin in vivo, embryonic myofiber formation is disturbed and the second wave of fetal myogenesis is largely abolished (*Hasty et al., 1993*; *Nabeshima et al., 1993*; *Venuti et al., 1995*). Notably, MRFs are able to induce phenotypical markers of skeletal muscle in permissive non-muscle cells (*Braun et al., 1990*; *Braun et al., 1989*; *Comai and Tajbakhsh, 2014*; *Davis et al., 1987*; *Edmondson and Olson, 1989*; *Weintraub et al., 1989*). Therefore, we examined whether MRFs regulate NE-MTOC formation during skeletal muscle differentiation and whether they are sufficient for NE-MTOC initiation in non-muscle cells.

NE-MTOC formation involves the localization of different MTOC proteins to the nuclear envelope. These include the PCM components pericentrin (PCNT), CDK5RAP2, and AKAP9 (also known as AKAP450) as well as γ-tubulin, key component of γ-tubulin ring complexes (γTuRCs) (*Bugnard et al., 2005*; *Espigat-Georger et al., 2016*; *Gimpel et al., 2017*; *Srsen et al., 2009*). At the centrosomal MTOC, PCNT, CDK5RAP2, and AKAP9 can interact with and recruit γTuRCs that in turn promote microtubule nucleation (*Teixido-Travesa et al., 2012*). At the nuclear envelope of myotubes, microtubule nucleation appears to specifically depend on AKAP9 (*Gimpel et al., 2017*) and γ-tubulin (*Bugnard et al., 2005*). Another protein localized to the nuclear envelope is PCM-1, an integral component of centriolar satellites, which contribute to recruiting proteins to the centrosome and proper organization of the centrosomal MTOC (*Prosser and Pelletier, 2020*). In myotubes, PCM-1 is required to recruit microtubule-associated motors to the nuclear envelope (*Espigat-Georger et al., 2016*). The localization of MTOC proteins at thstate nuclear envelope depends on the muscle-specific α-isoform of the outer nuclear membrane protein nesprin-1 (*Espigat-Georger et al., 2016*; *Gimpel et al., 2017*; *Holt, 2016*; *Randles et al., 2010*). Additionally, we recently discovered that – in cardiomyocytes – the large scaffold protein AKAP6 acts as an adapter between nesprin-1α and the MTOC proteins PCNT and AKAP9 (*Vergarajauregui et al., 2020*).

Based on loss- and gain-of-function experiments, quantitative analysis utilizing different cell types, and promoter studies, we show here that myogenin is required and sufficient for the formation of an NE-MTOC by controlling the expression of muscle- and NE-MTOC-specific isoforms of *Akap6* and *Syne1* that encodes nesprin-1α.

## Results

### Myogenin is required for MTOC protein localization to the nuclear envelope

To gain insight into the regulation of NE-MTOC establishment in skeletal muscle cells, we correlated the expression of MyoD and myogenin during mouse C2C12 myoblast differentiation with two key steps of NE-MTOC formation: (1) the expression of nesprin-1α, the nuclear envelope anchor for MTOC proteins (*Espigat-Georger et al., 2016*; *Gimpel et al., 2017*), and (2) the recruitment of PCM-1, the first MTOC protein localizing to the nuclear membrane (*Srsen et al., 2009*; *Zebrowski et al., 2015*). Immunofluorescence analyses of C2C12 cells 1 day after induction of differentiation revealed that 49.1% ± 7% of nuclei were MyoD+, 12% ± 0.9% were myogenin+, 7.8% ± 0.3% were nesprin-1α+, and 1.9% ± 0.3% were PCM-1+ (*Figure 1A–C*). Note that intermediate stages of PCM-1 nuclear envelope recruitment can be observed (*Figure 1—figure supplement 1A*), suggesting that PCM-1 recruitment occurs in a gradual manner. In our analysis, intermediate stages were rare (<2% of total PCM-1+ nuclei) and have therefore been included in the total percentage of PCM-1+ nuclei.

Prominent MyoD expression was detected in all cells that had upregulated nesprin-1α and recruited PCM1 to the nuclear envelope (*Figure 1A and B*), suggesting that MyoD-driven early differentiation is required for both steps of NE-MTOC formation. In contrast, the late differentiation factor myogenin was detected in all PCM1+ nuclei (*Figure 1B*) but only in 42% of nesprin-1+ nuclei (*Figure 1A and D*). Considering that (1) myogenin is a downstream target of MyoD (*Berkes and Tapscott, 2005*) and (2) nesprin-1α is anchoring MTOC proteins at the nuclear envelope, this suggested that, during NE-MTOC formation, myogenin might be either dispensable or required for a second step downstream of nesprin-1α.

To determine the role of myogenin in MTOC protein localization to the nuclear envelope, we depleted myogenin in C2C12 cells via siRNA and analyzed nesprin-1α expression as well as nuclear envelope localization of the MTOC proteins PCM-1, PCNT, and AKAP9 after 2 days of differentiation. The number of nesprin-1α+ nuclei was not significantly affected by myogenin depletion (*Figure 1E and F*). By contrast, knockdown of myogenin reduced the number of PCM-1+ nuclei from 11.6% ± 1.3% in control siRNA-treated cells to 2.3% ± 0.8% (*Figure 1E and F*) whereby PCM-1 retained a centrosomal localization pattern at highly nesprin-1α+ nuclei in myogenin-depleted cultures (*Figure 1—figure supplement 2*), although the pattern was less focused than in undifferentiated myoblasts. Similarly, PCNT+ nuclei were reduced from 6.9% ± 0.1% to 1.4% ± 0.1% and AKAP9+ nuclei showed a reduction from 6.8% ± 0.5% to 1.4% ± 0.1% (*Figure 1F*, *Figure 1—figure supplement 1B*). These data indicate that myogenin is required for MTOC protein localization to the nuclear envelope during skeletal muscle differentiation.

Considering that PCM-1 is (1) important for recruitment of other proteins to the centrosome (*Dammermann and Merdes, 2002*; *Prosser and Pelletier, 2020*) and (2) the first MTOC protein localizing to the nuclear envelope (*Srsen et al., 2009*; *Zebrowski et al., 2015*), we aimed to assess whether the loss of PCM-1 affects the recruitment of other MTOC proteins to the nuclear envelope. Depletion of PCM-1 reduced the percentage of C2C12 nuclei fully or partially positive for PCNT after 2 days of differentiation from 5.8% ± 1% to 0.8% ± 0.2% and 2.7% ± 0.9% to 0.5% ± 0.1%, respectively (*Figure 1G and H*). In contrast, PCM-1 depletion did not affect recruitment of AKAP9 (*Figure 1—figure supplement 1C*). Similar qualitative results have been described in myotubes (*Espigat-Georger et al., 2016*; *Gimpel et al., 2017*).

Collectively, our data suggest that the myogenin-independent early myogenic differentiation is sufficient to induce nesprin-1α expression, whereas myogenin regulates the nuclear envelope targeting of AKAP9 and PCM-1, which in turn recruits PCNT.

### Ectopic myogenin expression is sufficient to induce an NE-MTOC

All MRFs are able to 'transdifferentiate' permissive non-muscle cells with varying efficiency. They induce skeletal muscle markers such as the expression of contractile proteins or cell fusion into myotubes (*Braun et al., 1990*; *Braun et al., 1989*; *Davis et al., 1987*; *Edmondson and Olson, 1989*; *Weintraub et al., 1989*). However, NE-MTOC formation has never been analyzed in 'transdifferentiated' cells. To determine whether MRFs are sufficient to induce NE-MTOC formation in non-muscle cells, we ectopically expressed MyoD-GFP, myogenin-GFP, or GFP alone in mouse NIH3T3 fibroblasts and analyzed the localization of PCM-1. NIH3T3 cells transfected with GFP exhibited centrosomal

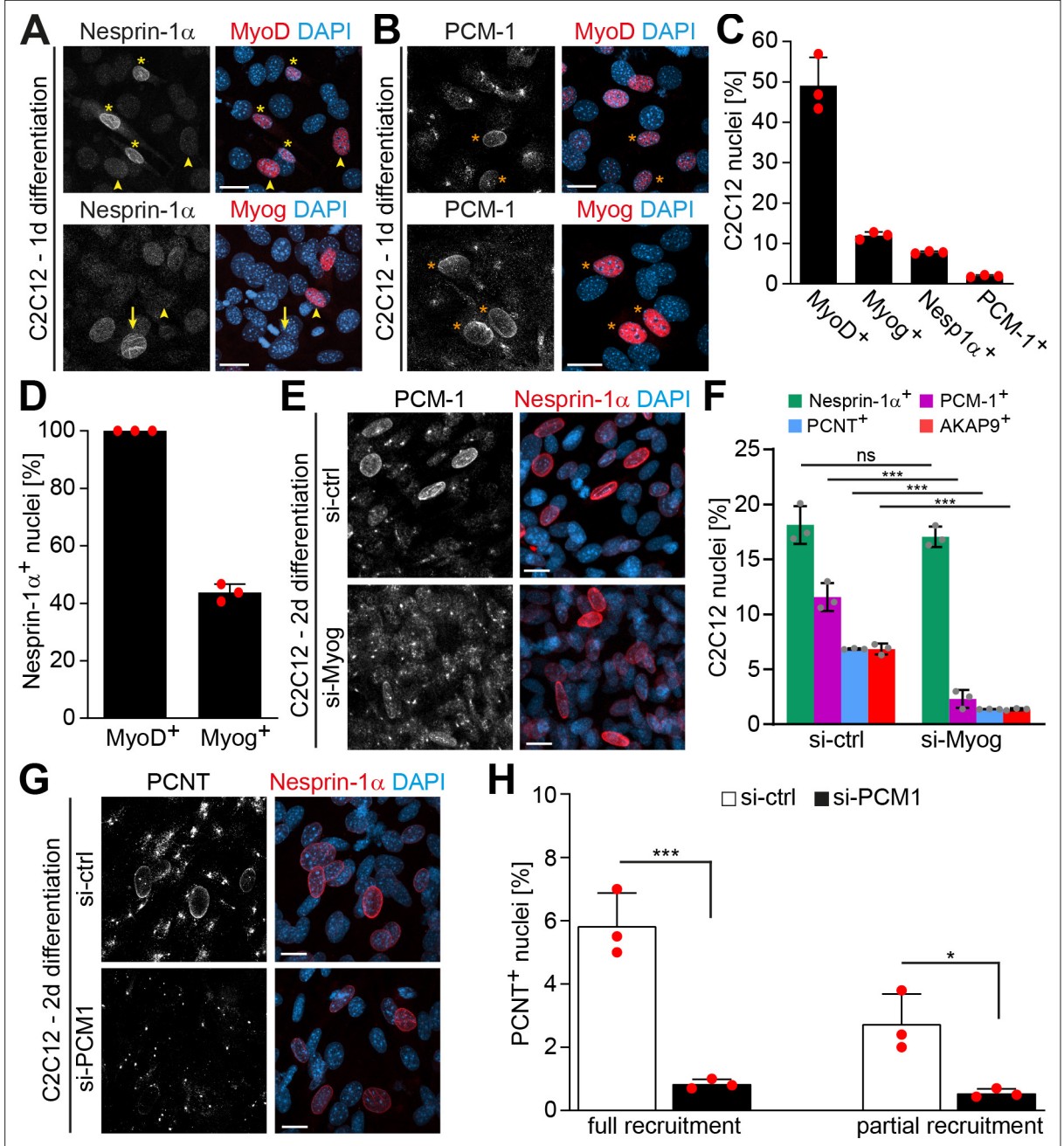

**Figure 1.** Myogenin is required for microtubule-organizing center (MTOC) protein localization to the nuclear envelope. (**A, B**) C2C12 cells were differentiated for 1 day and immunostained for the myogenic regulatory factors (MRFs) MyoD or myogenin (Myog) and nesprin-1α (**A**) or PCM-1 (**B**). Orange asterisks: MRF+/PCM-1+ nuclei; yellow asterisks: MRF+/nesprin-1α+ nuclei; arrowheads: MRF+/nesprin-1α- nuclei; arrows: MRF-/nesprin-1α+ nuclei. (**C**) Quantification of (**A**) and (**B**). (**D**) Quantification of MyoD and Myog in relation to nesprin-1α showing that not all nesprin-1α+ nuclei are myogenin+. (**E, F**) C2C12 myoblasts were transfected with negative control (si-ctrl) or *myogenin* siRNA (si-Myog) and differentiated for 2 days. Immunostaining (**E**) and subsequent quantification (**F**) shows that myogenin depletion affects nuclear envelope localization of PCM-1, PCNT, and AKAP9 but not of nesprin-1α. 95% CI of differences si-Myog vs. si-ctrl = –3.11% to 0.98% (nesprin-1α+), –11.31% to –7.22% (PCM-1+), –7.53% to 3.43% (PCNT+), and –7.52% to –3.42% (AKAP9+). (**G, H**) C2C12 myoblasts were transfected with si-ctrl or *Pcm1* siRNA (si-PCM1) and differentiated for 2 days. PCNT was detected by immunostaining (**G**) and subsequent quantification (**H**) showed that PCM-1 depletion reduces PCNT nuclei. 95% CI of differences si-PCM1 vs. si-ctrl = –6.6% to –3.4% (full), –3.78% to –0.59% (partial). Scale bars (**A, B, E, G**): 20 µm. Data (**C, D, F, H**) are represented as individual biological replicates (n = 3), together with mean ± SD. ns: p>0.05; *p<0.05; ***p<0.001.

The online version of this article includes the following figure supplement(s) for figure 1:

**Source data 1.** Underlying data for graphs in *Figure 1C, D and F* .

*Figure 1 continued*

**Figure supplement 1.** Intermediate stages of microtubule-organizing center (MTOC) protein localization to the nuclear envelope.

**Figure supplement 2.** PCM-1 exhibits a centrosomal localization pattern at nesprin-1+ nuclei in myogenin-depleted C2C12 cells.

PCM-1 localization, typical for proliferating cells (*Figure 2A*). By contrast, expression of MyoD-GFP and, surprisingly, myogenin-GFP induced nuclear envelope localization of PCM-1 in a subset of GFP+ cells (*Figure 2A and B*), suggesting that both MRFs are sufficient individually to induce localization of MTOC proteins to the nuclear envelope.

The results obtained upon MyoD-GFP expression are potentially explained by the fact that MyoD can activate myogenin transcription (*Berkes and Tapscott, 2005*). Analyzing endogenous myogenin expression in MyoD-GFP-expressing cells, we observed high myogenin levels in cells that had recruited PCM-1 to the nuclear envelope compared to MyoD-GFP-expressing cells where PCM-1 was absent from the nuclear envelope (*Figure 2—figure supplement 1A*). In addition, depletion of myogenin in NIH3T3 cells abrogated the MyoD-GFP-induced localization of PCM-1 to the nuclear envelope (*Figure 2—figure supplement 1B*). To confirm that MyoD regulates the expression of myogenin but not vice versa, we analyzed samples of differentiated C2C12 cultures depleted of MyoD or myogenin via RT-PCR (*Figure 2—figure supplement 1C*). While MyoD depletion also reduced *Myog* levels, *Myod1* levels were not detectably affected in myogenin-depleted cultures. Taken together, these results further argue that myogenin is required for the localization of MTOC proteins to the nuclear envelope.

In order to elucidate how myogenin – whose expression did not correlate with nesprin-1α in C2C12 cells (*Figure 1D*) – induces nuclear envelope MTOC formation, we generated stable NIH3T3 cell lines that express either mScarlet or myogenin-2A-mScarlet (MYOG-mScarlet) under control of a tetracycline-responsive promoter (Tet-ON). To induce myogenin expression and, potentially, MTOC protein recruitment to the nuclear envelope, we treated MYOG-mScarlet cells with doxycycline (Dox) for 3 days. Immunofluorescence analysis revealed that Dox treatment induced nuclear envelope localization of PCM-1, PCNT, and AKAP9 (*Figure 2C, E and G*). To better account for the dynamic of the recruitment process, we quantified the area of the nuclear envelope positive for each MTOC protein in mScarlet and MYOG-mScarlet cells after Dox treatment (*Figure 2D, F and H*). For this, we set an intensity threshold for the MTOC protein signal and quantified the percentage of pixels above this threshold inside a 1-µm-wide band around the nucleus (identified by DAPI signal). The signal at the centrosome, which localizes in close proximity to the nucleus in most cells, accounts for the nuclear envelope coverage in mScarlet cells. Quantification revealed that median PCM-1 nuclear envelope coverage increases from 12.5% in mScarlet cells to 30.3% in MYOG-mScarlet cells (*Figure 2D*). PCNT was recruited with an increase of the median coverage from 8.2% in mScarlet cells to 23.1% in MYOG-mScarlet cells (*Figure 2F*). AKAP9, which is essential for microtubule nucleation at the nuclear envelope (*Gimpel et al., 2017*), coverage increased only moderately from 3.6% to 6.2%, suggesting that AKAP9 recruitment is less efficient in MYOG-mScarlet cells compared to PCM-1 and PCNT (*Figure 2H*). However, ~13% of analyzed MYOG-mScarlet nuclei showed a higher coverage than the maximum observed in mScarlet cells. Together, these data indicate that myogenin is sufficient to induce nuclear envelope localization of MTOC proteins in non-muscle cells.

As nuclear envelope localization of MTOC proteins requires nesprin-1α in C2C12 cells (*Espigat-Georger et al., 2016*; *Gimpel et al., 2017*), we examined whether myogenin is able to induce nesprin-1α expression in Dox-treated MYOG-mScarlet cells. Dox treatment resulted in nesprin-1+ nuclei (*Figure 2I*), and RT-PCR analysis confirmed that the α-isoform transcript of *Syne1* is upregulated upon myogenin induction (*Figure 2J*). These data indicate that myogenin can induce nesprin-1α expression. Notably, chromatin immunoprecipitation (ChIP) sequencing data available through the ENCODE consortium (*Consortium et al., 2020*; *Yue et al., 2014*) predict myogenin as well as MyoD to bind a candidate regulatory element in the *Syne1* gene near the transcription start site of the α-isoform. To examine whether myogenin directly induces transcription of the *Syne1* α-isoform in MYOG-mScarlet cells by binding to the α-isoform-specific promoter in *Syne1*, we performed ChIP using an anti-myogenin antibody and an isotype control, followed by PCR for the ENCODE-predicted site. PCR amplification was successful from myogenin-precipitated DNA but not from IgG1 control (*Figure 2K*), indicating that myogenin binds the α-isoform promoter in *Syne1*. To further substantiate these results, we probed lysates of Dox-treated mScarlet cells and MYOG-mScarlet cells maintained

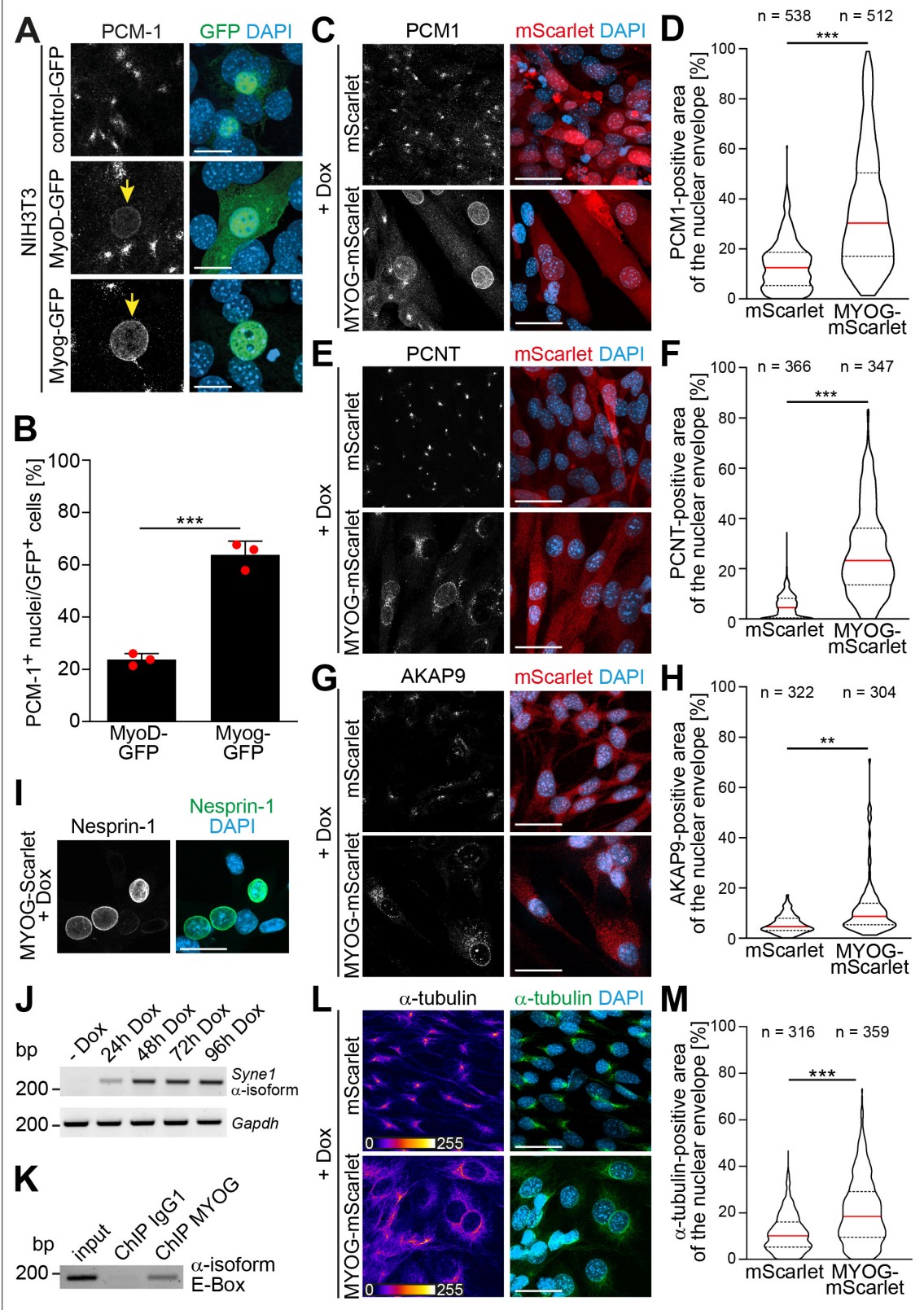

**Figure 2.** Myogenin expression is sufficient to induce nuclear envelope microtubule-organizing center (NE-MTOC) formation in non-muscle cells. (**A**) NIH3T3 fibroblasts were transfected with constructs encoding GFP, MyoD-GFP or myogenin-GFP (Myog-GFP). After three days, PCM-1 localization was assessed by immunostaining. Arrows indicate nuclei of transfected cells which have recruited PCM-1. Scale bars: 10 μm. (**B**) Quantification of (**A**) demonstrating that myogenin induces nuclear envelope localization of PCM-1 more efficiently than MyoD. Data are represented as individual biological

*Figure 2 continued on next page*

*Figure 2 continued*

replicates (n = 3), together with mean ± SD. ***: p < 0.001; 95% CI of difference Myog-GFP vs. MyoD-GFP = 30.99% to 49.22%. n = 3. (C-H) NIH3T3 Tet-ON mScarlet or MYOG-2A-mScarlet (MYOG-mScarlet) cells were treated with doxycycline (Dox) for three days. After immunostaining, nuclear envelope localization of PCM-1 (**C-D**), PCNT (**E-F**), and AKAP9 (**G-H**) was analyzed and quantified. Data are depicted as violin plots. Red line indicates the median, dotted lines indicate the 25% and 75% percentile. ***: p < 0.001. Scale bars: 20 µm (**I**) Immunostaining of MYOG-mScarlet cells treated with Dox for three days showing the presence of nesprin-1α+ nuclei. Scale bars: 20 µm (**J**) RT-PCR analysis of MYOG-mScarlet cells in the absence of Dox (-Dox) or treated with Dox for the indicated time points demonstrating that *nesprin-1α* is upregulated upon myogenin expression. *Gapdh* was used as equal input control. (**K**) ChIP-PCR analysis of Dox-treated MYOG-mScarlet cells using an anti-myogenin antibody or an IgG1 control showing that myogenin binds an E-box in the nesprin-1α promoter region. (**L-M**) Immunostaining of α-tubulin and subsequent quantification of nuclear envelope coverage after 30s of microtubule regrowth following cold-induced microtubule depolymerization in mScarlet or MYOG-mScarlet cells treated with Dox for three days. Data are depicted as violin plots. Red line indicates the median, dotted lines indicate the 25% and 75% percentile. ***: p < 0.001. Scale bars: 20 µm. N numbers indicate total number of analyzed nuclei pooled from three biological replicates.

The online version of this article includes the following figure supplement(s) for figure 2:

**Source data 1.** Underlying data for graphs in *Figure 2B, D, F and H*.

**Source data 2.** Raw files and uncropped gels for *Figure 2J*.

**Source data 3.** Raw files and uncropped gels for *Figure 2K*.

**Figure supplement 1.** Myogenin is required for MyoD-induced nuclear envelope recruitment of PCM-1.

**Figure supplement 1—source data 1.** Raw files and uncropped gels for *Figure 2—figure supplement 1C*.

**Figure supplement 2.** Myogenin targets are specifically precipitated in induced MYOG-mScarlet cells.

**Figure supplement 3.** Microtubule depolymerization in mScarlet and MYOG-mScarlet cells.

in Dox-free medium. Under both conditions, myogenin expression is not detectable. Additionally, we included an intronic region of *Syne1* as negative control as well as a promoter region of the known myogenin target *desmin* (*Londhe and Davie, 2011*) as positive control (*Figure 2—figure supplement 2*). Analysis revealed that the *Syne1* α-isoform promoter as well as the *desmin* promoter were precipitated using the myogenin antibody only in Dox-treated MYOG-mScarlet cells. Therefore, we conclude that myogenin binds the nesprin-1α promoter in *Syne1* and can induce expression of nesprin-1α in permissive cells.

Finally, we examined if the myogenin-induced recruitment of MTOC proteins converts the nuclear envelope to a functional MTOC. For this, we analyzed microtubule regrowth after cold-induced depolymerization in Dox-treated mScarlet and MYOG-mScarlet cells (*Figure 2L*, *Figure 2—figure supplement 3*). In mScarlet-expressing cells, microtubule regrowth was observed from the centrosome. In contrast, MYOG-mScarlet cells exhibited microtubule regrowth from the nuclear envelope to varying degrees. Quantification revealed that median nuclear envelope coverage increased from 10.1% in mScarlet cells to 18.4% in MYOG-mScarlet cells.

Collectively, these data demonstrate that myogenin is sufficient to induce NE-MTOC formation in NIH3T3 fibroblasts.

## Myogenin expression attenuates the centrosomal MTOC

In different cell types, it has been observed that ncMTOC formation is associated with attenuation of the centrosomal MTOC (*Leask et al., 1997*; *Muroyama et al., 2016*; *Nguyen et al., 2011*; *Yang and Feldman, 2015*; *Zebrowski et al., 2015*). Therefore, we examined if myogenin expression induces centrosome attenuation in MYOG-mScarlet fibroblasts. Dox stimulation resulted in a significant reduction of PCNT levels at centrioles in MYOG-mScarlet cells when compared to Dox-treated mScarlet cells (*Figure 3A and B*). In contrast, levels of Cep135, a centriole-associated protein, which does not relocalize to the nuclear envelope in muscle cells, did not change significantly upon myogenin induction (*Figure 3C and D*), indicating that myogenin affects centrosomal localization of PCM proteins but not of centriole-associated proteins. To test if myogenin attenuates MTOC activity at the centrosome, we analyzed centrosomal levels of the microtubule nucleating factor γ-tubulin. Induced MYOG-mScarlet cells displayed a significant reduction in centrosomal γ-tubulin levels compared to mScarlet control cells (*Figure 3E*). Analyzing microtubule regrowth, we observed that centrosomes still nucleated microtubules in MYOG-mScarlet cells that exhibited microtubule nucleation at the nuclear envelope (*Figure 3F*). However, α-tubulin signal at centrioles was less intense in MYOG-mScarlet cells compared to mScarlet cells, indicating a reduced MTOC activity (*Figure 3F*). Taken together, these

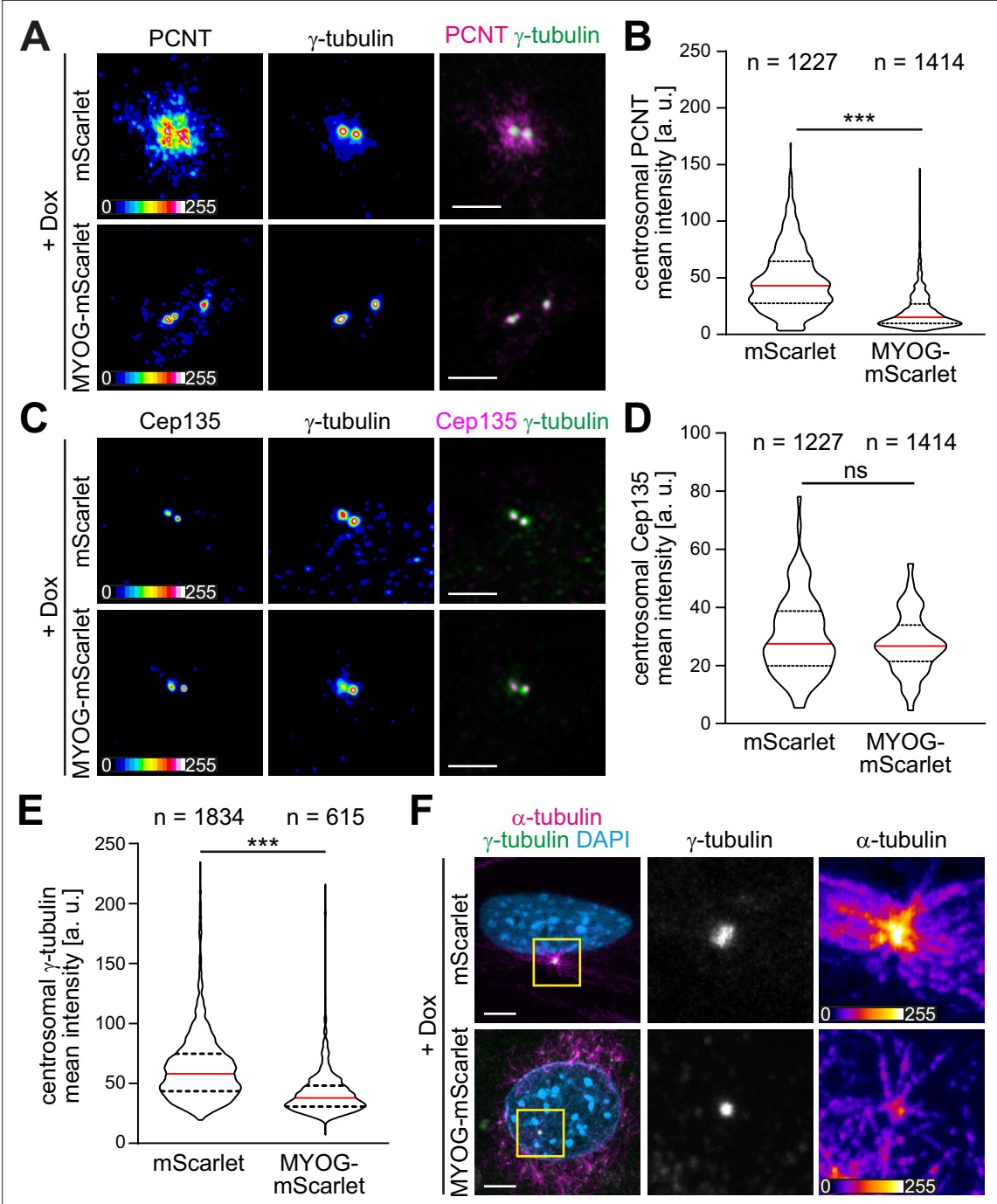

**Figure 3.** Myogenin expression attenuates the centrosomal microtubule-organizing center (MTOC). (**A–E**) mScarlet or MYOG-mScarlet cells were stimulated with doxycycline (Dox) for 3 days and PCNT (**A**), Cep135 (**C**), and γ-tubulin (**A, C**) were detected by immunostaining. Quantification shows that PCNT (**B**) and γ-tubulin (**E**) intensities at the centrosome are reduced upon myogenin induction while Cep135 intensity (**D**) is not significantly affected. Single-channel images of Pcnt, γ-tubulin, and Cep135 are false-colored to visualize different intensities. Data are shown as violin plots. The red line indicates the median, and dotted lines indicate the 25% and 75% percentile. ns: p>0.05; ***p<0.001. Scale bars = 5 μm. N numbers indicate the total number of analyzed centrioles (γ-tubulin foci) pooled from four biological replicates. (**F**) Immunostaining of α-tubulin and γ-tubulin in Dox-stimulated mScarlet or MYOG-mScarlet cells after 30 s of microtubule regrowth. Intensity-based color coding of α-tubulin shows that microtubule growth from centrioles is reduced after myogenin induction. Scale bars: 5 μm.

The online version of this article includes the following figure supplement(s) for figure 3:

**Source data 1.** Underlying data for graphs in *Figure 3B, D and E*.

data indicate that myogenin expression – in parallel to inducing ncMTOC formation – attenuates the centrosomal MTOC and that centrosomal and NE-MTOC can be active at the same time.

## AKAP6 is a potential mediator of myogenin-induced NE-MTOC formation

The myogenin depletion experiments suggested that the sole presence of nesprin-1α at the nuclear envelope does not allow efficient recruitment of MTOC components during muscle differentiation (*Figure 1F*). Thus, myogenin potentially contributes to the recruitment process by controlling the expression of proteins that are necessary for (1) inhibiting the localization of MTOC proteins to the centrosome, (2) targeting MTOC proteins to the nucleus, and/or (3) anchoring MTOC proteins to the nuclear envelope via nesprin-1α.

In order to identify candidates that act downstream of myogenin and mediate NE-MTOC formation, we integrated published myogenin ChIP-seq data (*Yue et al., 2014*) with RNA-seq data of C2C12 differentiation (*Doynova et al., 2017*; *Figure 4A*). Myogenin ChIP-seq data was obtained at four different time points (myoblasts, 24 hr differentiation, 60 hr differentiation, and 7 days differentiation; see Materials and methods for details), whereas the RNA-seq data set contained three time points (myoblasts as well as differentiating C2C12 cells at 3 days and 7 days of differentiation). PCM-1 nuclear envelope localization can already be observed 24 hr after induction of differentiation (*Figure 1A*), but the number of cells that differentiate and form an NE-MTOC significantly increases over time. Additionally, we assumed that genes required for the maintenance of the NE-MTOC in differentiated cells have to be actively transcribed. Therefore, we considered only those myogenin-binding sites in our analysis that were detected in the ChIP-seq data set at 24 hr, 60 hr, and 7 days of differentiation. The promoters of 2462 genes were bound by myogenin at these three time points (*Figure 4B*). We then intersected these 2462 genes with a list of 3800 genes, which were upregulated in the RNA-seq data set at both 3 days and 7 days of differentiation when compared to proliferating myoblasts (*Figure 4B*). This intersection yielded a list of 748 potential direct myogenin target genes (*Figure 4B*). Considering that skeletal muscle cells and cardiomyocytes (myogenin-negative) both express nesprin-1α and exhibit an NE-MTOC, we hypothesized that NE-MTOC formation in both cell types is controlled by similar mechanisms. Thus, we assessed whether any of the 748 target genes are upregulated during rat heart development from embryonic day 15 to postnatal day 3, the developmental window in which NE-MTOCs form in cardiomyocytes (*Zebrowski et al., 2015*). For this purpose, we utilized a microarray-derived temporal expression data set spanning rat heart development (*Patra et al., 2011*). This strategy helped to further reduce the number of candidate genes to 107 myogenin targets that potentially mediate NE-MTOC formation (*Supplementary file 1*). As our previous data suggested that nesprin-1α alone does not allow efficient recruitment of MTOC proteins to the nuclear envelope, we first focused on candidates that potentially cooperate with nesprin-1α in anchoring MTOC proteins to the nuclear envelope. To this end, we utilized Gene Ontology analysis to identify candidates that are annotated to localize at the nuclear envelope (*Figure 4A*). Four genes matched the Gene Ontology cellular component search terms 'nuclear membrane' and 'nuclear envelope': *Akap6*, *Dmpk*, *Rb1cc1*, and *Tmem38a*. Previous studies indicated that myogenin directly binds and activates the promoter of *Akap6* (*Lee et al., 2015*), which encodes the large scaffold A-kinase anchoring protein (AKAP) 6 (also known as mAKAP). AKAP6 has been described to localize to the nuclear envelope of cardiomyocytes through interaction with the N-terminal spectrin domains of nesprin-1α and to act as a signaling hub by assembling signaling proteins such as protein kinase A, ryanodine receptor, phosphodiesterase 4D3, and phospholipase C (*Kapiloff et al., 1999*; *Pare et al., 2005*; *Passariello et al., 2015*; *Ruehr et al., 2003*). Furthermore, proximity labeling indicated that AKAP6 is an interactor of nesprin-1α in C2C12 myotubes (*Gimpel et al., 2017*) and a recent study in our lab identified AKAP6 as a key organizer of the NE-MTOC in cardiomyocytes (*Vergarajauregui et al., 2020*). Taken together, these data identify AKAP6 as a potential mediator of myogenin-induced NE-MTOC formation.

AKAP6 occurs in two isoforms: the brain-specific α-isoform and the β-isoform, which is predominantly expressed in heart and skeletal muscle (*Michel et al., 2005*). We first determined if myogenin binds the β-isoform promoter of *Akap6* in MYOG-mScarlet fibroblasts and if *Akap6β* expression is induced in these cells upon Dox treatment. We could amplify an E-box-containing region of the β-isoform promoter after ChIP using an anti-myogenin antibody (*Figure 4C*). This result was confirmed by

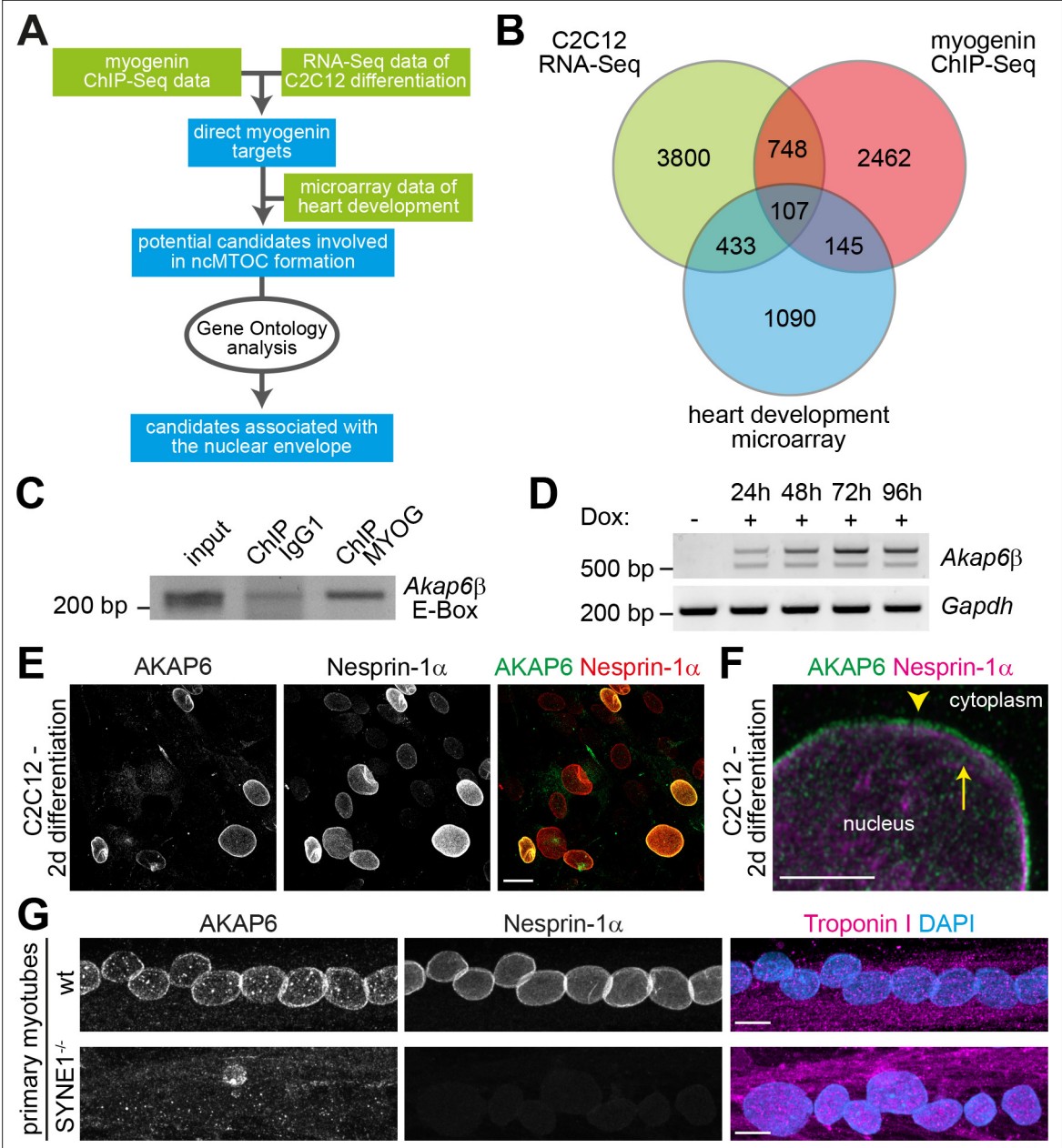

**Figure 4.** The nesprin-1α interaction partner AKAP6 is a potential mediator of myogenin-induced nuclear envelope microtubule-organizing center (NE-MTOC) formation. (**A**) Scheme illustrating the bioinformatics workflow used to identify potential myogenin downstream candidates. (**B**) Venn diagram depicting the numbers of genes matching criteria for the individual data sets and for intersection of data sets. Criteria for myogenin ChIP-seq data (red): Genes where myogenin binding was detected at the promoter region; criteria for C2C12 RNA-seq data (green) and for microarray data of rat heart development (blue): upregulated genes. (**C**) ChIP-PCR analysis of doxycycline (Dox)-treated MYOG-mScarlet cells using an anti-myogenin antibody or an IgG1 control showing that myogenin binds an E-box in the *Akap6β* promoter region. (**D**) RT-PCR analysis of MYOG-mScarlet cells in the absence of Dox (-Dox) or treated with Dox for the indicated time points demonstrating that *Akap6β* is upregulated upon myogenin expression. The two bands for *Akap6β* derive from alternative splicing of the first exon of *Akap6β*, which results in an ~200 bp insertion in the 5' untranslated region. *Gapdh* was used as equal input control. Please note that the same samples and *Gapdh* control were used as in ***Figure 2J***. (**E**) C2C12 cells were differentiated for 2 days, and immunostaining shows that all AKAP6+ nuclei are also nesprin-1α+. Scale bar: 20 μm. (**F**) High-resolution Airyscan image of (**E**). Arrowhead indicates AKAP6 localized at the cytoplasmic side of nesprin-1α signal. Arrow marks nesprin-1α that is localized at the nuclear side of AKAP6 signal. Scale bar: 0.5 μm. (**G**) Myoblasts from healthy donors (wt) and from patients carrying a mutation in the *SYNE1* gene (SYNE1-/-) were differentiated for 4 days. Immunostaining analysis showed that loss of nesprin-1α is associated with loss of AKAP6 from the nuclear envelope in differentiated myotubes (troponin I). Scale bars: 10 μm.

The online version of this article includes the following figure supplement(s) for figure 4:

*Figure 4 continued on next page*

*Figure 4 continued*

**Source data 1.** Raw files and uncropped gels for *Figure 4C*.

**Source data 2.** Raw files and uncropped gels for *Figure 4D*.

**Figure supplement 1.** The myogenin antibody specifically precipitates the *Akap6β* promoter region in induced MYOG-mScarlet cells.

**Figure supplement 2.** Quantification of nesprin-1α and AKAP6 signal at the nuclear envelope.

qPCR analysis of the ChIP samples revealing that the *Akap6β* promoter region is specifically enriched after precipitation with the myogenin antibody in Dox-treated MYOG-mScarlet samples but not in Dox-treated mScarlet or untreated MYOG-mScarlet samples (*Figure 4—figure supplement 1*). Consistently, RT-PCR analysis showed that the β-isoform of *Akap6* is upregulated after Dox stimulation (*Figure 4D*). Collectively, these data indicate that, in fibroblasts, myogenin can bind the *Akap6* β-isoform promoter and induce AKAP6 expression.

To determine whether nesprin-1α is – similar to the situation in cardiomyocytes – involved in AKAP6β localization to the nuclear envelope of skeletal muscle cells (*Pare et al., 2005*), we analyzed the expression pattern of AKAP6 (refers to the β isoform if not specified) and nesprin-1α in C2C12 cells. Immunofluorescence analysis at 2 days differentiation showed that all AKAP6+ nuclei were nesprin-1α+ (*Figure 4E*). In addition, high-resolution microscopy suggested that AKAP6 mainly localizes at the cytoplasmic side of nesprin-1α (*Figure 4F*, *Figure 4—figure supplement 2*). It has been reported that the C-terminus of nesprin-1α is inserted into the outer nuclear membrane, whereas the N-terminus extends into the cytoplasm (*Apel et al., 2000*; *Zhang et al., 2001*).

To test if nesprin-1α is required to anchor AKAP6 to the nuclear envelope, patient-derived myoblasts carrying a mutation in the *SYNE1* gene (23560 G>T causing a premature stop and loss nesprin-1α expression) and myoblasts of healthy donors were differentiated into myotubes and AKAP6 localization was compared (*Figure 4G*). Whereas nesprin-1α and AKAP6 localized to the nuclear envelope of myotubes from healthy donors, expression of nesprin-1α and nuclear membrane localization of AKAP6 were abolished in myotubes carrying the *SYNE1* mutation. Taken together, these data indicate that AKAP6 localization to the nuclear envelope in differentiated skeletal muscle cells depends on nesprin-1α.

## AKAP6 is required for NE-MTOC formation and maintenance

To examine the role of AKAP6 in NE-MTOC formation, we performed siRNA-mediated depletion experiments in differentiating C2C12 cells and MYOG-mScarlet fibroblasts. In C2C12 cultures differentiated for 2 days, 12.3% ± 0.6% of nuclei were AKAP6+ and 10.4% ± 1.2% of nuclei were PCM-1+ (*Figure 5A and B*). Importantly, AKAP6 was found at all nuclei that had recruited PCM-1. Transfection of differentiating C2C12 cultures with *Akap6* siRNA significantly reduced the number of AKAP6+ nuclei from 12.3% ± 0.6% to 4.8% ± 0.1% and the number of PCM-1+ nuclei from 10.4% ± 1.2% to 3.4% ± 0.3% (*Figure 5B*) but had no effect on nesprin-1α localization (*Figure 5—figure supplement 1*). Correspondingly, treatment of Dox-induced MYOG-mScarlet with *Akap6* siRNA decreased median nuclear envelope coverage by PCM-1 and PCNT from 22.6% to 9% and 18.7% to 5.5%, respectively (*Figure 5C–E*). Median coverage of AKAP9 was only moderately affected (7.2–6.5%) but nuclei showing more than ~18% AKAP9 coverage were completely lost after AKAP6 depletion (*Figure 5F*). To examine if AKAP6 promotes nuclear envelope recruitment by forming a complex with MTOC proteins, we performed co-immunoprecipitation experiments using an anti-AKAP6 antibody. We could co-precipitate PCM-1 from MYOG-mScarlet lysates but not from mScarlet lysate (*Figure 5G*). These data indicate that AKAP6 is required for the localization of MTOC proteins to the nuclear envelope, in part by forming a protein complex including PCM-1.

To examine whether the recruitment of MTOC proteins to the nuclear envelope is the reason for the attenuation of the centrosomal MTOC in MYOG-mScarlet fibroblasts, we analyzed centrosomal levels of PCNT and γ-tubulin in AKAP6-depleted or nesprin-1α-depleted cells (*Figure 5—figure supplement 2*). We did not observe an increase of centrosomal PCNT or γ-tubulin levels in AKAP6- or nesprin-1α-depleted cultures, indicating that the main mechanism for centrosome attenuation is not the competition with the NE-MTOC.

To determine if AKAP6 is required for maintaining MTOC protein localization at the nuclear envelope, we transfected C2C12 cultures enriched for myotubes with AKAP6 siRNA (*Figure 5H and I*).

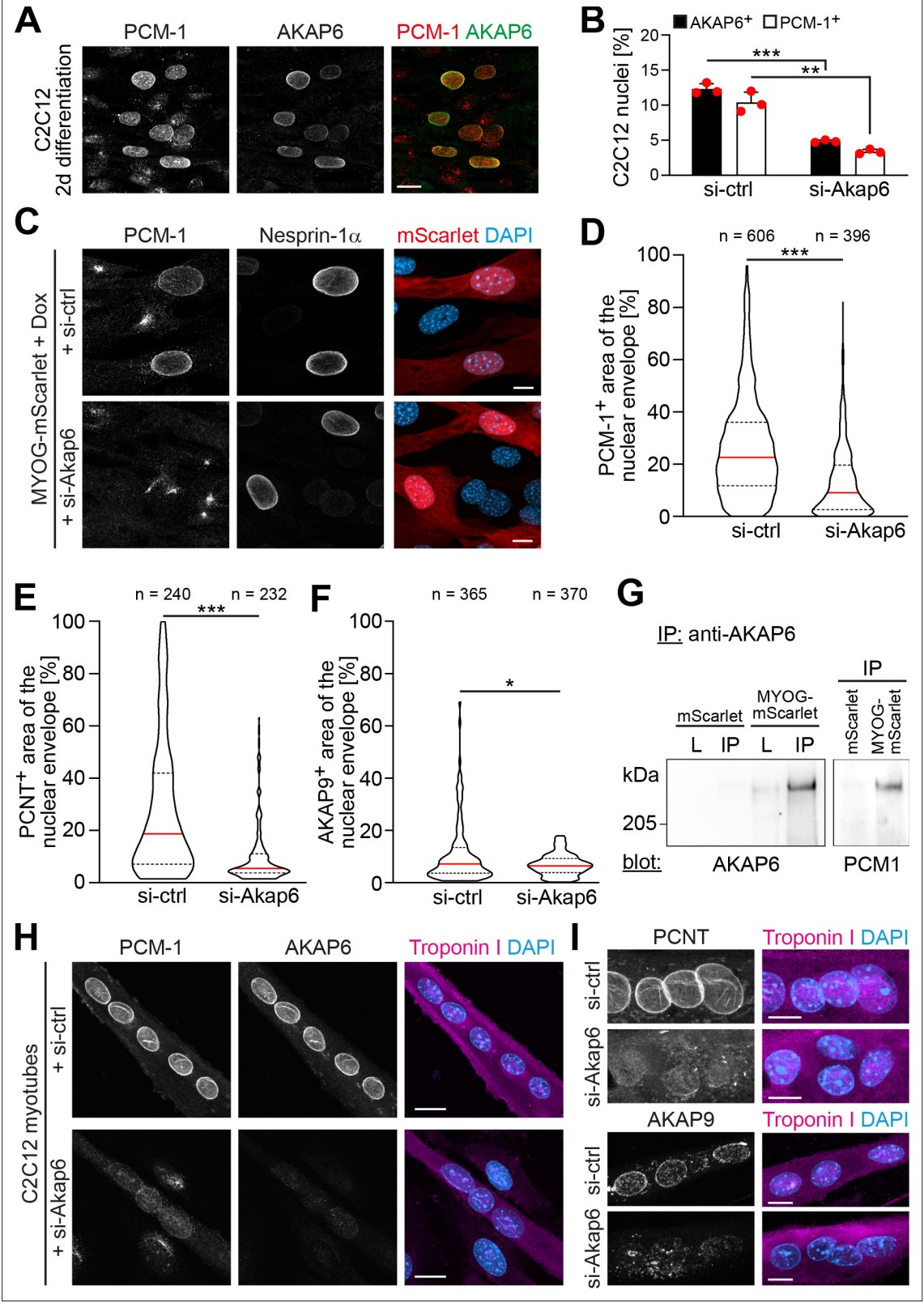

**Figure 5.** AKAP6 is required for the nuclear envelope localization of microtubule-organizing center (MTOC) proteins. (**A**) C2C12 cells were differentiated for 2 days. Immunostaining shows that all PCM-1+ nuclei are also AKAP6+. (**B**) Quantification of AKAP6+ and PCM-1 nuclei in C2C12 cells treated with negative control (si-ctrl) or *Akap6* (si-Akap6) siRNA after 2 days of differentiation indicates that AKAP6 is required for nuclear envelope localization of PCM-1. Data are represented as individual biological replicates (n = 3), together with mean ± SD. 95% CI = 6.21% to 8.74%; 95% CI = 4.63% to 9.43%.

*Figure 5 continued on next page*

*Figure 5 continued*

(**C**) MYOG-mScarlet cells were treated with si-ctrl or si-Akap6 and subsequently treated with doxycycline (Dox) for 3 days. Image analysis revealed that myogenin-induced localization of PCM-1 to the nuclear envelope is AKAP6-dependent. (**D**) Quantification of (**C**). (**E, F**) Quantification of PCNT (**E**) and AKAP9 (**F**) nuclear coverage in Dox-stimulated MYOG-mScarlet cells treated with si-ctrl or si-Akap6. (**G**) Co-immunoprecipitation (IP) of PCM-1 from MYOG-mScarlet but not from mScarlet lysate (**L**) using an anti-AKAP6 antibody. (**H, I**) Enriched C2C12 myotubes (troponin I) were transfected with si-ctrl or si-Akap6 and immunostaining demonstrates that AKAP6 is required for maintaining nuclear envelope localization of PCM-1 (**H**) as well as PCNT and AKAP9 (**I**). Scale bars (**A, H**) 20 µm, (**C, I**) 10 µm. Data (**D–F**) are shown as violin plots. The red line indicates the median, and dotted lines indicate the 25% and 75% percentile. N numbers indicate the total number of analyzed nuclei pooled from three biological replicates. *p<0.05.; **p<0.01; ***p<0.001.

The online version of this article includes the following figure supplement(s) for figure 5:

**Source data 1.** Underlying data for graphs in *Figure 5B, D–F*.

**Source data 2.** Raw files and uncropped blots for *Figure 5G*.

**Figure supplement 1.** AKAP6 depletion does not affect nesprin-1α.

**Figure supplement 2.** AKAP6 or nesprin-1α depletion does not prevent centrosome attenuation in MYOG-expressing cells.

**Figure supplement 3.** MyoD induces nesprin-1α and AKAP6.

Depletion of AKAP6 resulted in the loss of PCM-1, AKAP9, and PCNT from the nuclear envelope in myotubes.

Taken together, our data demonstrate that AKAP6 is required for recruiting MTOC proteins to the nuclear envelope as well as maintaining nuclear envelope localization of MTOC proteins in myotubes, most likely by acting as an adaptor between MTOC proteins and the nuclear membrane anchor nesprin-1α.

## MyoD can induce AKAP6 expression via myogenin

Similar to myogenin, ectopic expression of MyoD was sufficient to induce PCM-1 localization to the nuclear envelope (*Figure 2A and B*). Consistently, a more detailed analysis of MyoD-GFP-transfected NIH3T3 cells revealed that nesprin-1 and AKAP6 expression is induced in these cells as well (*Figure 5—figure supplement 3*). As previous depletion experiments indicated that MyoD induces PCM-1 localization to the nuclear envelope via myogenin (*Figure 2—figure supplement 1*), we analyzed nesprin-1 and AKAP6 in MyoD-GFP-transfected cells treated with *Myog* siRNA. Analysis revealed that the percentage of nesprin-1+ nuclei in GFP+ cells was not affected by myogenin depletion (*Figure 5—figure supplement 3*), which is consistent with our findings in C2C12 cells (*Figure 1F*). However, the percentage of AKAP6+ as well as PCM-1+ nuclei was reduced upon myogenin depletion. This further shows that MyoD-induced MTOC protein localization to the nuclear envelope depends on the induction of myogenin.

## AKAP6 is required for NE-MTOC function

The NE-MTOC has been described to be required for correct positioning and distribution of nuclei in multinucleated myotubes via two different mechanisms: (1) PCM-1 enables the recruitment of the dynein regulator p150glued and other motor proteins to the nuclear envelope and promotes alignment of nuclei (*Espigat-Georger et al., 2016*), and (2) AKAP9-dependent nucleation of microtubules from the nuclear envelope contributes to the spreading of nuclei throughout the cell body (*Gimpel et al., 2017*; *Figure 6A*). Our results indicate a potential role for AKAP6 in both aspects of nucleus positioning as it is required for PCM-1 and AKAP9 to localize to the nuclear envelope. Analyzing the positioning and distribution of nuclei in enriched C2C12 myotubes 2 days after siRNA-mediated depletion of AKAP6, we found that AKAP6 depletion reduced the number of myotubes with aligned nuclei (66.0% ± 7.5% to 34.7% ± 7.6%) and increased the number of myotubes with overlapping nuclei (26.0% ± 6.5% to 58.3% ± 9.6%), compared to control myotubes (*Figure 6B and C*). This indicates that AKAP6 is required for proper alignment and spreading of nuclei in myotubes.

Next, we aimed to confirm that the observed nuclei mispositioning in AKAP6-depleted cells is due to aberrant microtubule nucleation and motor protein recruitment at the nuclear envelope. Immunofluorescence analysis showed that 2 days post siRNA transfection the microtubule network organization was similar in AKAP6-depleted and control myotubes showing the typical organization of microtubules in longitudinal arrays (*Figure 6—figure supplement 1A*). Similar results have been

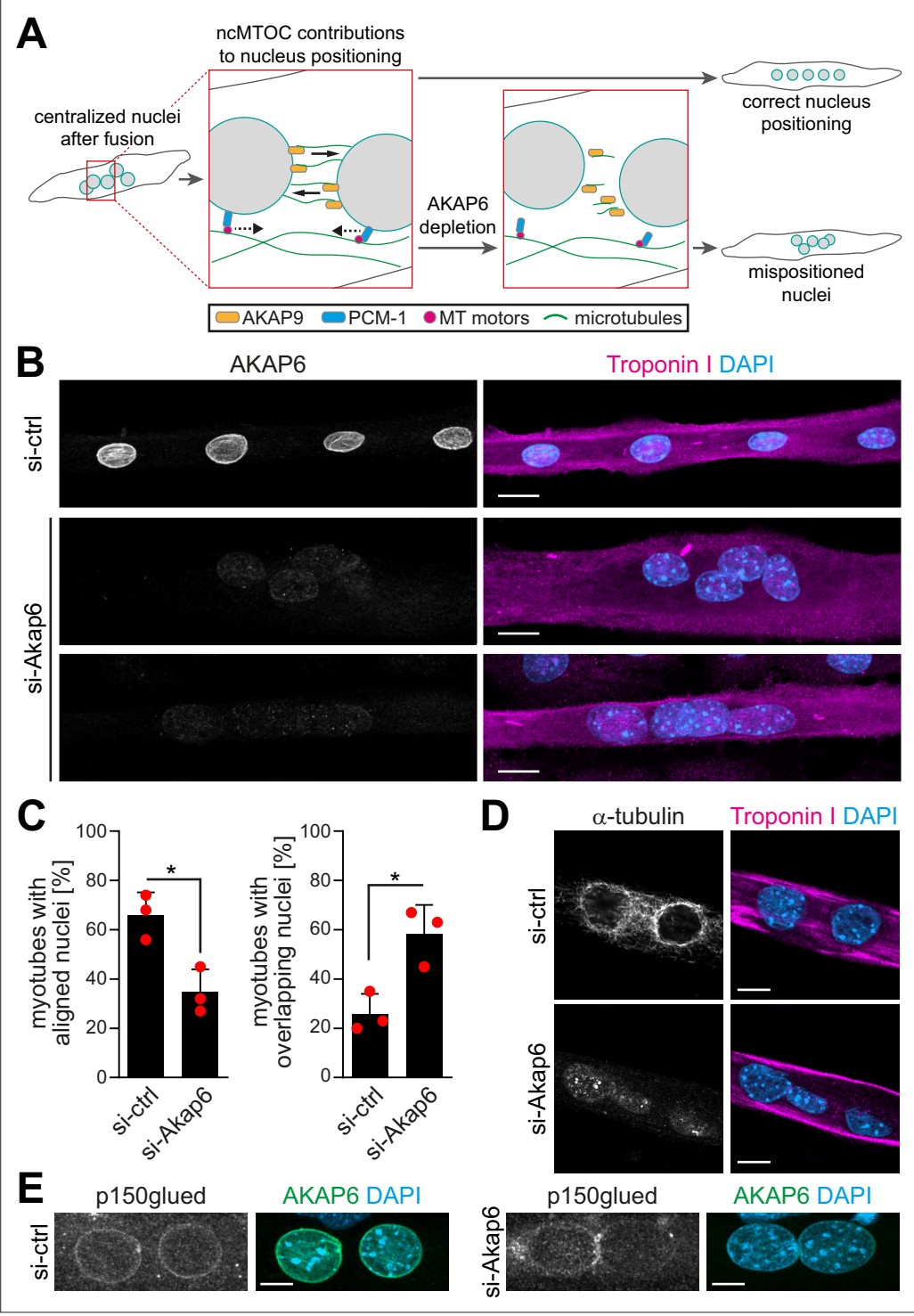

**Figure 6.** AKAP6 is required for correct nuclear positioning in myotubes. (**A**) Scheme illustrating the role of the nuclear envelope microtubule-organizing center (NE-MTOC) in myonuclear positioning and the potential impact of AKAP6 depletion. (**B**) Enriched C2C12 myotubes (troponin I) were transfected with negative control (si-ctrl) or *Akap6* (si-Akap6) siRNA. The upper si-Akap6 panel shows a representative image of a myotube with misaligned nuclei, and the lower si-Akap6 panel shows nuclei overlapping inside a myotube. (**C**) Quantification of (**B**). Data are represented as individual biological replicates (n = 3), together with mean ± SD. *p<0.05, 95% CI of difference si-Akap6 vs. si-ctrl = 10.42% to 52.25% (left graph); 95% CI = 9.65% to 55.02% (right graph). (**D**) Enriched C2C12 myotubes (troponin I) were transfected with si-ctrl or si-Akap6 and subsequently subjected to a nocodazole-based

*Figure 6 continued on next page*

*Figure 6 continued*

microtubule (α-tubulin) regrowth assay. Image analysis showed that AKAP6 depletion abrogated microtubule nucleation at the nuclear envelope. (**E**) Enriched C2C12 myotubes (troponin I) transfected with si-ctrl or si-Akap6 were immunostained for the dynein regulator p150glued. Image analysis showed that AKAP6 depletion reduces p150glued signal at the nuclear envelope. Scale bars (**B**) 20 µm, (**D**) 10 µm, and (**E**) 5 µm.

The online version of this article includes the following figure supplement(s) for figure 6:

**Source data 1.** Underlying data for graphs in *Figure 6C*.

**Figure supplement 1.** AKAP6 depletion does not affect steady state microtubule organization but reduces detyrosinated microtubules.

---

obtained previously when depleting nesprin-1α (*Espigat-Georger et al., 2016*). Yet, we observed that AKAP6 depletion resulted in a reduced intensity of detyrosinated (i.e., stable) microtubules compared to control myotubes (*Figure 6—figure supplement 1B*). To test if AKAP6 depletion affects the nucleation of new microtubules, we assessed microtubule regrowth after nocodazole-induced depolymerization in C2C12 myotubes. In myotubes treated with control siRNA, microtubules regrew from the nuclear envelope and to a lesser extent from cytoplasmic loci (*Figure 6D*). In AKAP6-depleted myotubes, microtubule regrowth from the nuclear envelope was abolished (*Figure 6D*). This suggests that AKA6 depletion impairs spreading of myonuclei by preventing microtubule growth from the nuclear envelope. Next, we analyzed the localization of p150glued (also known as DCTN1). In control siRNA-treated myotubes, p150glued localized at the nuclear envelope (*Figure 6E*). AKAP6 depletion resulted in a reduction of p150glued at the nuclear envelope of C2C12 myotubes (*Figure 6E*), suggesting that the reduced number of myotubes with aligned nuclei is due to impaired dynein activation. Collectively, these data demonstrate that AKAP6 is required for the function of the NE-MTOC in skeletal muscle cells.

## Myogenin-induced isoforms of nesprin-1 and AKAP6 are sufficient for MTOC protein recruitment

To examine if myogenin specifically induces expression of isoforms that are associated with the NE-MTOC in skeletal muscle, we performed ChIP on MYOG-mScarlet cell lysate using an anti-myogenin antibody and assessed the abundance of isoform-specific promoter regions of *Syne1* and *Akap6* in the precipitated DNA. For this, we performed qPCR using primer pairs targeting myogenin consensus binding sites (i.e., E-boxes) in regions predicted by ENCODE data (*Consortium et al., 2020*) to be associated with myogenin binding (*Figure 7A and C*). We found that the amount of template corresponding to the promoter region upstream of the *Syne1* α-isoform transcript (nesprin-1α2) was 4.5-fold higher than the promoter region of the long *Syne1* isoform (nesprin-1-giant) (*Figure 7B*). Similarly, the promoter region upstream of the *Akap6β* transcript was threefold enriched compared to the promoter region of the Akap6α transcript (*Figure 7D*). This indicates that myogenin preferentially binds promoter regions of *Syne1* and *Akap6* isoforms that are involved in NE-MTOC formation.

To test if the preferential binding is associated with an increased activation of transcription of specific isoforms, we constructed vectors with putative promoter regions of the α- or β-isoform of *Akap6* as well as with promoter regions of the giant- or α-isoform of *Syne1* located directly upstream of a luciferase coding sequence. These promoter constructs were then co-transfected into human HEK293T cells together with GFP or myogenin-GFP. Co-transfection of myogenin with the *Akap6* β-isoform promoter construct increased luciferase activity 10.9-fold, while co-transfection with the *Akap6* α-isoform promoter construct did not show a significant increase compared to GFP-transfected control (*Figure 7E*). Similarly, we observed a 21.7-fold increase in activity when myogenin was co-transfected with the *Syne1* α-promoter construct but only a mild 2.7-fold increase after co-transfection with the promoter construct of the giant isoform of *Syne1* (*Figure 7F*). These results indicate that myogenin preferentially induces transcription of the *Syne1* α-isoform and the *Akap6* β-isoform.

Finally, we examined whether the myogenin-induced isoforms of nesprin-1 and AKAP6 are sufficient to recruit MTOC proteins in the absence of myogenin. For this, we expressed nesprin-1α-mCherry alone or together with AKAP6β-GFP in undifferentiated, myogenin-negative myoblasts. In nesprin-1α-mCherry-transfected cells, PCM-1 did not localize to the nuclear envelope (*Figure 7G*). In contrast, co-transfection of nesprin-1α-mCherry and AKAP6β-GFP was sufficient to recruit PCM-1

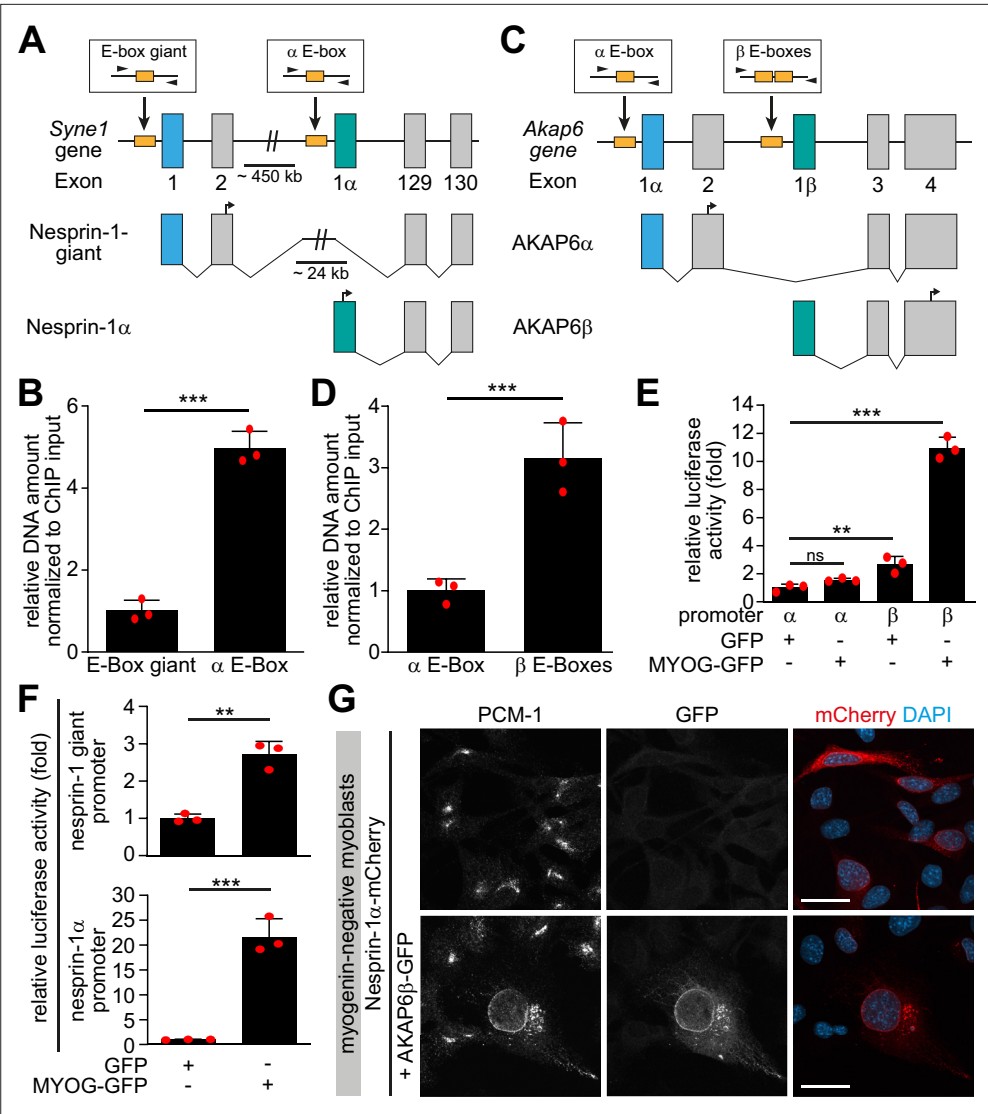

**Figure 7.** Myogenin preferentially induces microtubule-organizing center (MTOC)-associated isoforms of *Syne1* and *Akap6*. (**A, C**) Schematic representation of the murine *Syne1* (**A**) and *Akap6* (**C**) gene and derived transcripts. Exons are indicated by gray rectangles and the first exon of each transcript is marked by color. E-boxes (myogenin consensus sites) inside putative promoters are indicated as yellow boxes and small black arrows mark the primers used for qPCR. (**B, D**) Myogenin chromatin immunoprecipitation (ChIP) from doxycycline (Dox)-stimulated MYOG-mScarlet cells followed by qPCR for the indicated E-boxes shows that myogenin preferentially binds the promoter regions upstream of *Syne1* α-isoform and *Akap6* β-isoform transcripts. (**E, F**) Luciferase assay testing the activity of the indicated *Akap6* (**E**) or *Syne1* (**F**) promoters in the presence of GFP or myogenin-GFP (MYOG-GFP). (**G**) Overexpression of nesprin-1α-mCherry alone or together with AKAP6β-GFP in undifferentiated (myogenin-negative) C2C12 myoblasts. Co-expression of nesprin-1α and AKAP6β is sufficient for nuclear envelope recruitment of endogenous PCM-1. Scale bars: 20 μm. Data (**B, D–F**) are represented as individual biological replicates (n = 3), together with mean ± SD. ns: p>0.05; **p<0.01; ***p<0.001.

The online version of this article includes the following figure supplement(s) for figure 7:

**Source data 1.** Underlying data for graphs in *Figure 7B, D–F*.

**Figure supplement 1.** Ectopic co-expression of nesprin-1α and AKAP6β is not sufficient for microtubule-organizing center (MTOC) function at the nuclear envelope.

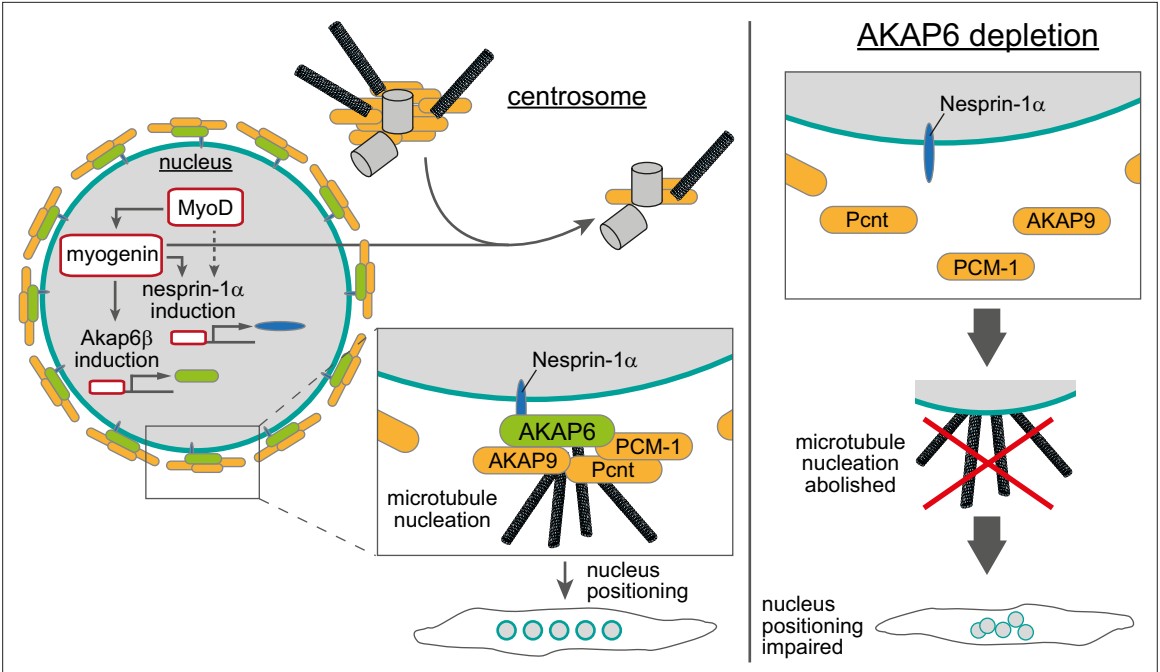

**Figure 8.** Schematic overview of the role of myogenin and AKAP6 in nuclear envelope microtubule-organizing center (NE-MTOC) formation. Myogenin induces expression of AKAP6β that connects MTOC proteins like PCNT, AKAP9, and PCM-1 to the nuclear membrane protein nesprin-1α, whose expression can be induced by myogenin as well as MyoD. Depletion, overexpression, and co-immunoprecipitation experiments suggest that AKAP6β acts as an adapter between MTOC proteins and nesprin-1α. Yet, other proteins might be involved and the here presented protein complex at the nuclear envelope is hypothetical. At the same time, myogenin is sufficient to attenuate centrosomal MTOC function. AKAP6-dependent anchoring of MTOC proteins as well as microtubule nucleation from the nuclear envelope are required for correct positioning of nuclei inside differentiating myotubes.

to the nuclear envelope (*Figure 7G*). To test whether co-expression of nesprin-1α and AKAP6β is sufficient to convert the nuclear envelope to a functional MTOC, we performed microtubule regrowth experiments (*Figure 7—figure supplement 1*). Microtubules regrew from the centrosome and in the cytoplasm, but significant regrowth from the nuclear envelope was not observed.

Taken together, our results demonstrate that myogenin specifically induces transcription of isoforms that are (1) required for the NE-MTOC in differentiated skeletal muscle cells and (2) sufficient to recruit MTOC proteins to the nuclear envelope in cells with a centrosomal MTOC.

## Discussion

We conclude that the myogenic transcription factor myogenin controls NE-MTOC formation and that myogenin-induced AKAP6β expression is one of the central molecular components required for NE-MTOC formation (*Figure 8*). This conclusion is supported by our findings that (1) myogenin is required for the localization of the MTOC proteins PCM-1, PCNT, and AKAP9 to the nuclear envelope in differentiating muscle cells, (2) ectopic myogenin expression is sufficient to promote the formation of an NE-MTOC in fibroblasts, and (3) the myogenin-induced isoforms AKAP6β and nesprin-1α are required and sufficient for the recruitment of MTOC proteins to the nuclear envelope.

Formation of ncMTOCs has been associated with cellular differentiation (*Sanchez and Feldman, 2017*), but a direct regulation of ncMTOC formation by particular differentiation pathways in vertebrate cells has remained elusive. Our results demonstrate that myogenin, which is an essential regulator of terminal differentiation, drives NE-MTOC formation in mammalian cells. This shows that terminal differentiation factors can control in vertebrates the switch of dominant MTOC localization from the centrosome to non-centrosomal sites. Notably, MTOC formation at the nuclear envelope occurs also in cells that lack myogenin or cell-type-specific transcriptional master regulators of terminal differentiation, such as cardiomyocytes or osteoclasts (*Kronebusch and Singer, 1987*; *Mulari et al., 2003*; *Zebrowski et al., 2015*). A recent study from our lab demonstrated that AKAP6β orchestrates the

assembly of the NE-MTOC in cardiomyocytes and osteoclasts (*Vergarajauregui et al., 2020*), validating our findings in skeletal muscle. Therefore, it would be important in future studies to identify transcription factors that bind to the *Akap6* β-isoform as well as *Syne1* α-isoform promoters and regulate the switch from centrosomal to NE-MTOC in these cell types. For this purpose, promising cardiomyocyte and osteoclast 'transdifferentiation' tools using multiple transcription factors are available (*Chang et al., 2019*; *Ieda et al., 2010*; *Klose et al., 2019*; *Yamamoto et al., 2015*). Additionally, it appears important to determine in future experiments the mechanisms underlying the preferential binding of myogenin to the isoform-specific promoters, considering the abundance of myogenin binding sites (E-boxes) throughout the genome. Notably, isoform upregulation or switching might be a general mechanism that contributes to MTOC regulation during differentiation. This assumption is supported by the recent identification of a spermatid-specific isoform of centrosomin, the *Drosophila* orthologue of CDK5RAP2, which can induce ncMTOC formation at mitochondria (*Chen et al., 2017*). Moreover, a non-centrosomal isoform of ninein contributes to neuronal differentiation (*Zhang et al., 2016*) and a shorter isoform of PCNT is upregulated in differentiating cardiomyocytes (*Zebrowski et al., 2015*).

Both MyoD and myogenin induced expression of nesprin-1α and AKAP6 as well as the nuclear envelope localization of PCM-1. The depletion experiments in differentiating C2C12 cells and in MyoD-transfected fibroblasts indicate that myogenin is required for the localization of MTOC proteins to the nuclear envelope via AKAP6 expression. While myogenin can induce nesprin-1α expression in fibroblasts, it is dispensable for nesprin-1α upregulation during C2C12 differentiation. Together, these findings suggest a model of NE-MTOC formation during C2C12 differentiation in which MyoD induces nesprin-1α as well as myogenin, which is then required to induce AKAP6 expression allowing recruitment of MTOC proteins to the nuclear envelope.

The centrosomal MTOC is attenuated in differentiated muscle cells (*Becker et al., 2020*). Consistently, we found that ectopic myogenin expression in fibroblasts resulted in reduced MTOC protein levels at centrosomes as well as attenuated centrosomal microtubule regrowth. Depletion of nesprin-1α or AKAP6 in this system – abolishing the localization of MTOC proteins to the nuclear envelope – did not result in obvious reactivation of the centrosomal MTOC, indicating that centrosome attenuation is not due to competition with the NE-MTOC. Furthermore, differentiating C2C12 cells in which myogenin was depleted maintained PCM-1 in a centriolar satellite-like pattern, albeit this pattern was less focused. Taken together, these results suggest that myogenin attenuates the centrosome during muscle differentiation independently of inducing NE-MTOC formation.

While site-specific anchor proteins for ncMTOCs, such as nesprin-1α, have been identified (*Espigat-Georger et al., 2016*; *Gimpel et al., 2017*; *Lechler and Fuchs, 2007*; *Meng et al., 2008*), it remained unclear how MTOC proteins are connected to these site-specific anchors. Previously, it has been reported that overexpression of nesprin-1α in undifferentiated myoblasts is sufficient to recruit an ectopically expressed centrosomal targeting domain of PCNT and AKAP9 (i.e., the PACT domain) as well as minor amounts of endogenous PCM-1 to the nuclear envelope in a subset of transfected cells (*Espigat-Georger et al., 2016*; *Gimpel et al., 2017*). Here, we show that myogenin induced the expression of the large scaffold protein AKAP6, which we prove to be essential for NE-MTOC formation and maintenance, most likely by connecting MTOC proteins to nesprin-1α. Myogenin preferentially binds and activates the putative promoters of AKAP6β and nesprin-1α isoforms, which are known to be upregulated in differentiated muscle cells (*Kapiloff et al., 1999*; *Michel et al., 2005*; *Randles et al., 2010*). Importantly, ectopic co-expression of AKAP6β and nesprin-1α, but not nesprin-1α alone, was sufficient to recruit endogenous MTOC proteins in the absence of myogenin.

While co-expression of AKAP6β and nesprin-1α induced nuclear envelope recruitment of PCM-1 in undifferentiated myoblasts, microtubule regrowth in these cells was readily observed at the centrosome but not at the nuclear envelope. This indicates that AKAP6β and nesprin-1α alone are not sufficient to generate an active NE-MTOC. As described above, NE-MTOC formation and centrosome attenuation appear to be independently regulated by myogenin. Thus, one explanation for the absence of NE-MTOC activity in the co-expression experiment might be that recruitment of MTOC proteins to the nuclear envelope is not efficient enough to compete with the non-attenuated centrosomal MTOC for microtubule nucleation factors. However, it appears also possible that additional myogenin-downstream mechanisms (e.g., induction of specific microtubule nucleators) are needed to activate the NE-MTOC after MTOC proteins have been recruited.

Our results indicate an important role of AKAP6-dependent NE-MTOC function in nucleus positioning in skeletal myotubes in vitro. While nucleus positioning is a frequent feature of human myopathies (*Jungbluth and Gautel, 2014*; *Madej-Pilarczyk and Kochański, 2016*), the specific role of the NE-MTOC in these pathologies remains largely elusive. Yet, mutations in the nesprin-1 gene *SYNE1* have been described in EDMD patients, pointing towards a potential role of NE-MTOC defects in this pathology (*Zhang et al., 2007*). Fully elucidating the mechanisms of NE-MTOC formation in vivo and the specific contributions of nuclear envelope-originated microtubules to the different aspects of myonuclear positioning will clarify if and how NE-MTOC defects contribute to human myopathies. In addition to nucleus positioning, microtubules help to maintain nuclear architecture in myotubes (*Wang et al., 2015*) and cardiomyocytes (*Heffler et al., 2020*) and also regulate contractility (*Chen et al., 2018*). While perinuclear microtubules have been identified to be specifically important for nuclear architecture, the significance of the NE-MTOC in this context remains unclear. Precise targeting of the NE-MTOC via AKAP6 appears a promising strategy to elucidate the role of nuclear envelope-generated microtubules in maintaining nuclear architecture as well as regulating contractility. Finally, amplified and/or hyperactive centrosomes act as oncogene-like factors (*Arnandis et al., 2018*; *Godinho and Pellman, 2014*; *Godinho et al., 2014*; *Levine et al., 2017*; *LoMastro and Holland, 2019*). Therefore, it is important to better understand mechanisms that control MTOC activity. Notably, ectopic expression of myogenin in fibroblasts did not only induce NE-MTOC formation but also attenuated the centrosomal MTOC. In addition, myogenin expression in fibroblast was only inducing in a subset of cells an NE-MTOC. Thus, our cellular systems combined with our bioinformatics approach provide new opportunities to tackle future key questions of MTOC formation such as: What factors increase or decrease efficiency of NE-MTOC induction? What post-transcriptional processes contribute to NE-MTOC establishment? How is centrosome attenuation achieved?

In summary, our findings suggest that key differentiation factors can control the switch from centrosomal MTOC to ncMTOC and cell-type-specific adaptor proteins are required to connect MTOC proteins to anchor proteins at non-centrosomal sites. Conclusively, our study (1) contributes to a better understanding of the striated muscle NE-MTOC, (2) presents a mechanistic framework that may be applicable to ncMTOC formation in other cell types and tissues, and (3) provides a cellular system to elucidate further molecular mechanisms inducing the switch from centrosomal to ncMTOCs.

# Materials and methods

**Key resources table**

| Reagent type (species) or resource | Designation | Source or reference | Identifiers | Additional information |
|---|---|---|---|---|
| Cell line (*Mus musculus*) | C2C12 | ATCC | Cat# CRL-1772, RRID: CVCL_0188 | Myoblast cell line |
| Cell line (*M. musculus*) | NIH3T3 | ATCC | Cat# CRL-1658, RRID:CVCL_0594 | Fibroblast cell line |
| Cell line (*Homo sapiens*) | HEK293T | ATCC | Cat# PTA-4488, RRID:CVCL_0045 | |
| Cell line (*H. sapiens*) | Human myoblast healthy donor | (*Holt, 2016*; *Mamchaoui et al., 2011*); Institute de Myologie, Paris | | |
| Cell line (*H. sapiens*) | Human myoblast patient-derived | (*Holt, 2016*; *Mamchaoui et al., 2011*); Institute de Myologie, Paris | | Mutation in the *SYNE1* gene (23560 G>T causing a premature stop and loss nesprin-1α expression) |
| Antibody | Anti-PCM1 (rabbit polyclonal) | Santa Cruz | Cat# sc-67204, RRID:AB_2139591 | WB (1:500), IF (1:200) |
| Antibody | Anti-AKAP6 (rabbit polyclonal) | Sigma-Aldrich | Cat# HPA048741, RRID:AB_2680506 | WB (1:2000), IP/IF (1:500) |

*Continued on next page*

*Continued*

| Reagent type (species) or resource | Designation | Source or reference | Identifiers | Additional information |
|---|---|---|---|---|
| Antibody | Anti-PCM1 (mouse monoclonal) | Santa Cruz | Cat# sc-398365, RRID:AB_2827155 | IF (1:200) |
| Antibody | Anti-nesprin-1 (MANNES1E) (mouse monoclonal) | G.Morris (*Randles et al., 2010*) | | IF (1:50) |
| Antibody | Anti-myogenin (mouse monoclonal) | Santa Cruz | Cat# sc-12732, RRID:AB_627980 | IF (1:500) |
| Antibody | Anti-MyoD1 (mouse monoclonal) | Millipore | Cat# MAB3878, RRID:AB_2251119 | IF (1:500) |
| Antibody | Anti-tubulin (rat monoclonal) | Sigma-Aldrich | Cat# T9026, RRID:AB_477593 | IF (1:500) |
| Antibody | Anti-Troponin I (goat polyclonal) | Abcam | Cat# ab56357, RRID:AB_880622 | IF (1:500) |
| Antibody | Anti-γ-tubulin (mouse monoclonal) | Santa Cruz | Cat# sc-51715, RRID:AB_630410 | IF (1:100) |
| Antibody | Anti-AKAP9 (rabbit polyclonal) | Sigma-Aldrich | Cat# HPA026109, RRID:AB_1844688 | IF (1:200) |
| Antibody | Anti-Pericentrin (rabbit polyclonal) | BioLegend | Cat# PRB-432C, RRID:AB_291635 | IF (1:1000) |
| Recombinant DNA reagent | psiCHECK-2 vector | Promega | Cat# C8021; GenBank Accession Number AY535007 | |
| Recombinant DNA reagent | peGPF-N1 | Clontech | Cat# 6085-1; GenBank Accession Number U55762 | |
| Recombinant DNA reagent | psPAX2 | D.Trono (Addgene) | Addgene plasmid #12260; RRID: Addgene_12260 | Lentiviral packaging plasmid |
| Recombinant DNA reagent | pMD2.G | D.Trono (Addgene) | Addgene plasmid #12259; RRID: Addgene_12259 | Lentiviral VSV-G envelope plasmid |
| Recombinant DNA reagent | pLenti CMVtight Blast DEST (w762-1) | E.Campeau (Addgene) | Addgene plasmid #26434; RRID: Addgene_26434 | Lentiviral transfer plasmid for Tet-ON system |
| Recombinant DNA reagent | pLenti CMV rtTA3 Hygro (w785-1) | E.Campeau (Addgene) | Addgene plasmid #26730; RRID: Addgene_26730 | Lentiviral transfer plasmid for Tet-ON system |
| Recombinant DNA reagent | mScarlet | D.Gadella (Addgene) | Addgene plasmid #85042; RRID: Addgene_85042 | |
| Sequence-based reagent | *MyoD1* cold fusion cloning forward | This paper | Cloning PCR primer | gggatccaccggtcgccac catggagcttctatcgccgcc |
| Sequence-based reagent | *MyoD1* cold fusion cloning reverse | This paper | Cloning PCR primer | tcctcgcccttgctcacc ataagcacctgataaatcgcat |
| Sequence-based reagent | *Myog* cold fusion cloning forward | This paper | Cloning PCR primer | gggatccaccggtcgccaccatggagctgtatgagacatc |
| Sequence-based reagent | *Myog* cold fusion cloning reverse | This paper | Cloning PCR primer | tcctcgcccttgctcaccatgttgggcatggtttcgtctg |
| Sequence-based reagent | *myogenin* siRNA | Integrated DNA technologies | Cat# mm.Ri.Myog.13.1 | AAUAAAGACUGGUUGCUAUCAAAAA |

*Continued on next page*

*Continued*

| Reagent type (species) or resource | Designation | Source or reference | Identifiers | Additional information |
|---|---|---|---|---|
| Sequence-based reagent | *Akap6* siRNA | Thermo Fischer Scientific | Cat# 4390771 s108732 | GGACUACAUCAAGAACGAATT |
| Sequence-based reagent | *Syne1* siRNA | Integrated DNA technologies | Cat# mm.Ri.Syne1.13.1 | AACUAGAGCUUAUCAACAAACAGTA |
| Sequence-based reagent | *Pcm1* siRNA | Integrated DNA technologies | Cat# mm.Ri.Pcm1.13.1 | AGUCAGAUUCUGCAACAUGAUCUTG |
| Sequence-based reagent | Negative control (si-ctrl) siRNA | Integrated DNA technologies | Cat# 51-01-14-04 | Non-targeting |
| Commercial assay or kit | Dual-Luciferase Reporter Assay System | Promega | Cat# E1910 | |
| Chemical compound, drug | Doxycycline hydrochloride | Sigma-Aldrich | Cat# D3447 | |
| Chemical compound, drug | Bovine fetuin | Thermo Fisher Scientific | Cat# 10344026 | |
| Chemical compound, drug | EGF Recombinant Human Protein | Thermo Fisher Scientific | Cat# PHG0311 | |
| Chemical compound, drug | FGF-Basic (AA 10-155) Recombinant Human Protein | Thermo Fisher Scientific | Cat# PHG0026 | |
| Chemical compound, drug | Insulin-Transferrin-Selenium-Sodium Pyruvate (ITS-A) (100X ) | Thermo Fisher Scientific | Cat# 51300044 | |
| Software, algorithm | Fiji software package | http://fiji.sc/ | RRID:SCR_002285 | |
| Software, algorithm | Bioconductor | http://www.bioconductor.org/ | RRID: SCR_006442 | |
| Other | Skeletal Muscle Differentiation Medium | PromoCell | Cat# C-23061 | |
| Other | Horse serum | Thermo Fisher Scientific | Cat# 16050122 | |

Further information and requests for resources and reagents should be directed to and will be fulfilled by the lead contact, Felix Engel (felix.engel@uk-erlangen.de).

## Cell lines, differentiation, and doxycycline stimulation

Cell types were authenticated as follows: human myoblasts and C2C12, myotube formation; NIH3T3, morphology; HEK293, efficiency in protein production. Note that the identity of NIH3T3 and HEK293T cells is not essential for this study. All cell lines were mycoplasma-free (tested every 12 months).

Reagents used for cell culture are listed in the Key resources table. All cells used in this study were cultured at 37 °C in a humidified atmosphere containing 5% $CO_2$. Growth medium for C2C12, NIH3T3, Hela, and Hct116 consisted of high glucose DMEM supplemented with GlutaMAX containing 10% FBS, 1 mM sodium pyruvate, 100 U/ml penicillin, and 100 µg/ml streptomycin. C2C12 cells were maintained at 50% confluence to preserve differentiation capacity. For differentiation, cells were cultured to 90% confluence and subsequently changed to differentiation medium high glucose DMEM with GlutaMAX containing 0.5% FBS and insulin, transferrin, selenium, sodium pyruvate solution (1:1000 of 100× ITS-A).

Human myoblasts from a healthy control or from a congenital muscular dystrophy patient carrying a homozygous nonsense mutation within the *SYNE1* gene (nucleotide 23560 G>T) were immortalized by Kamel Mamchaoui and Vincent Mouly (Center for Research in Myology, Paris, France) as previously described via transduction with retrovirus vectors expressing hTERT and Cdk4 (*Holt, 2016*; *Mamchaoui et al., 2011*). Growth medium consisted of DMEM supplemented with GlutaMAX and DMEM 199 in a 4:1 ratio containing 20% FBS, 25 µg/ml bovine fetuin, 5 ng/ml recombinant human EGF, 0.5 mg/ml recombinant human FGF-basic, 5 µg/ml recombinant insulin, 0.2 µg/ml dexamethasone, and 50 µg/ml gentamicin (*Gimpel et al., 2017*). To induce differentiation, immortalized myoblasts were grown to ~90% confluence and then changed to Skeletal Muscle Differentiation Medium (PromoCell) containing 50 µg/ml gentamicin. For immunofluorescence analysis of immortalized myoblasts, glass coverslips were coated with Matrigel diluted 1:100 in DMEM.

## Myotube enrichment

C2C12 cells were differentiated in 6-well plates or 10 cm dishes for 4–5 days as described above. To preferentially detach myotubes, cells were washed two times with PBS and treated with pre-warmed 0.0125% Trypsin/EDTA solution (0.25% Trypsin/EDTA stock diluted in PBS) for ~2 min at room temperature. Detachment of myotubes was constantly monitored by phase contrast microscopy. After sufficient myotube detachment was observed, Trypsin/EDTA solution was carefully aspirated and a myotube-enriched suspension was collected by rinsing the plates five times with normal growth medium. Enriched myotubes were then plated on glass coverslips coated with 25 µg/ml fibronectin in PBS for >45 min at 37 °C. After 24 hr incubation at 37 °C, myotube cultures were subjected to siRNA treatment and/or microtubule regrowth assays.

## MRF plasmids construction

*Myod1* and *Myog* coding sequences were obtained by PCR using cDNA from C2C12 cells differentiated for 2 days. The cDNAs were then cloned into the peGFP-N1 backbone by Cold Fusion Cloning (System Biosciences, Cat# MC010B-1) following the manufacturer's instruction. Positive clones were identified by restriction digest and Sanger sequencing.

## Luciferase plasmids construction

Candidate regulatory elements associated with myogenin binding in *Syne1* and *Akap6* genes were identified from ENCODE data accessed through the SCREEN web interface (https://screen.wenglab.org/). Potential promoter regions were amplified from genomic DNA obtained from NIH3T3 cells using the primers listed in *Supplementary file 2*. After amplification, promoter fragments were cloned in front of the *Renilla* luciferase ORF (*hRluc*) into the psiCHECK-2 vector using NEBuilder HiFi DNA Assembly Master Mix (New England Biolabs, Cat# E2621L) according to the manufacturer's instructions.

## Plasmid transfections

Plasmid transfection into NIH3T3 cells was carried out with 500 ng DNA per well of a 24-well plate using 1 µl Lipofectamine LTX (Thermo Fisher Scientific, Cat# 15338100) according to the manufacturer's instructions. Transfection complexes were formed by incubating DNA with Lipofectamine LTX in Opti-MEM for 20 min at room temperature.

For transfection of luciferase constructs in HEK293T cells, 250 ng luciferase vector and 250 ng myogenin-eGFP or eGFP control plasmid were used per well of a 24-well plate. Transfection complexes were assembled by incubating DNA with PEI MAX (Polysciences, Cat# 24765-1) in a 1:3 ratio in Opti-MEM for 20 min at room temperature.

## siRNA transfections

Cells were transfected using 2 µl Lipofectamine RNAiMAX reagent (Thermo Fisher Scientific, Cat# 13778150) and 40 nM final siRNA concentration per well of a 24-well plate. Transfection complexes were formed by incubating siRNA with RNAiMAX in Opti-MEM for 20 min at room temperature. C2C12 cells were transfected 24 hr prior to induction of differentiation (~50% confluence) and enriched C2C12 myotubes were transfected 24 hr after re-plating. NIH3T3 were transfected with siRNA 48 hr after plasmid transfection.

## Luciferase assay

Luciferase activity was measured in Centro XS³ LB 960 96-well plate reader luminometer (Berthold-Tech, #50-6860) using the Dual-Luciferase Reporter Assay System according to the manufacturer's instructions. In brief, HEK293T cells were harvested 48 hr after transfection in passive lysis buffer and stored at –80 °C until measurement. Activities of firefly luciferase (*hluc+*, internal control) and *Renilla* luciferase (promoter activation) were measured sequentially for each sample. Values of *Renilla* luciferase activity were normalized to those of firefly luciferase for each measurement.

## Production of lentiviral vectors

The lentiviral packaging plasmid psPAX2 and the VSV-G envelope plasmid pMD2.G were gifts from Didier Trono (psPAX2: Addgene plasmid #12260; http://n2t.net/addgene:12260; RRID: Addgene_12260; pMD2.G: Addgene plasmid #12259; http://n2t.net/addgene:12259; RRID: Addgene_12259). The transfer plasmids pLenti CMVtight Blast DEST (w762-1) and pLenti CMV rtTA3 Hygro (w785-1) used for creating tetracycline-inducible cell lines were gifts from Eric Campeau (w762-1: Addgene plasmid #26434; http://n2t.net/addgene:26434; RRID: Addgene_26434; w785-1: Addgene plasmid #26730; http://n2t.net/addgene:26730; RRID: Addgene_26730). The coding sequences of human myogenin (gift from Matthew Alexander & Louis Kunkel; Addgene plasmid #78341; http://n2t.net/addgene:78341; RRID: Addgene_78341) and mScarlet (*Bindels et al., 2017*; gift from Dorus Gadella; Addgene plasmid #85042; http://n2t.net/addgene:85042; RRID: Addgene_85042) were cloned into w762-1 by NEBuilder HiFi DNA Assembly according to the manufacturer's instructions using the primers indicated in *Supplementary file 2*.

To produce lentiviral vectors, psPAX2, pMD2.G, and the desired transfer plasmid were transfected in a 1:1:2 ratio into HEK293T cells using PEI MAX. Supernatant containing lentiviral vectors was harvested 72 hr after transfection, filtered through a 0.45 µm filter and aliquots were snap frozen.

## Generation of stable cell lines

Lentiviral vector aliquots were rapidly thawed at 37 °C and diluted in tetracycline-free growth medium containing 10 µg/ml Polybrene (Sigma-Aldrich, Cat# 107689). Cells were transduced overnight in 6-well plates using 1 ml of diluted lentiviral vector. The following morning, medium was refreshed and cells were selected for transgene integration 72 hr after transduction. We first generated cells expressing a reverse tetracycline activator (rtTA3), which were subsequently transduced with lentiviral vectors carrying the desired transgene to express mScarlet or MYOG-2A-mScarlet under control of a tetracycline-responsive element (TRE). After selection, Tet-ON cell lines were used for 20 passages.

## RT-PCR

RNA was isolated using a column-based RNA purification kit. For production of cDNA, 1 µg of RNA was reverse transcribed using Oligo (dT) 12-18mer primers and M-MLV Reverse Transcriptase (Sigma-Aldrich #M1302) according to the manufacturer's instructions. For PCR, ~20 ng of cDNA were used with Redtaq master mix (Genaxxon #M3029) and products were analyzed using agarose gel electrophoresis.

## Immunoprecipitation

Cells were harvested in lysis buffer containing 1% NP-40, 150 mM NaCl, 10% glycerol, 5 mM EDTA, 20 mM Tris-HCl (pH 7), and EDTA-free protease inhibitor cocktail (cOmplete, Roche # 11873580001). After 30 min incubation on ice, samples were sonicated and lysates were cleared by centrifugation at 16,000×g for 16 min at 4 °C. For immunoprecipitation, 0.5 µg anti-AKAP6 antibody/mg of total protein were added to the lysate and incubated overnight at 4 °C on a rotor. Subsequently, antibody complexes were purified by incubation of lysate with Protein A Sepharose beads (Merck, GE17-5138-01) for 3 hr rotating at 4 °C. Beads were washed three times with cold lysis buffer for 5 min and proteins were eluted from beads by incubation in 2× NuPAGE LDS sample buffer at 95 °C for 5 min. Lysates and immunoprecipitated samples were analyzed by SDS-PAGE (4–12% NuPAGE Novex Bis-Tris gels) under reducing conditions and transferred to a nitrocellulose membrane by wet transfer at 350 mA and <60 V for 1.5 hr in 1× transfer buffer (25 mM Tris-HCl, pH 7.5, 192 mM glycine, 0.1% SDS, 10% methanol). The membrane was then blocked with 5% BSA in TBS-T (1× TBS, 0.05% Tween-20) and incubated with primary antibodies against AKAP6 or PCM-1.

## ChIP-qPCR

Protein and DNA were cross-linked by fixing cells for 10 min at 37 °C using 1% formaldehyde in culture medium. Cross-linking was quenched by adding 125 mM glycine and gently agitating the cells for 5 min at room temperature. Then, cells were harvested in ice-cold PBS, centrifuged at 1000×g for 5 min at 4 °C, and the resulting pellet was lysed in ChIP lysis buffer (1% SDS, 10 mM EDTA, 50 mM Tris-HCl pH 8.1) for 30 min on ice. After lysis, samples were sonicated for 30 cycles consisting of 20 s sonication and 30 s pause inside an ice bath. Sonicated samples were centrifuged at 4 °C for 30 min at 10,000×g. A small aliquot of the supernatant was saved as input control and the remaining supernatant was subjected to immunoprecipitation. Samples were diluted 1:5 in RIPA buffer and incubated overnight at 4 °C with 1 µg/ml myogenin antibody (Santa Cruz Biotechnology, #sc-12732 X) or IgG1 isotype control (Thermo Fisher, # 16471482). Protein G agarose beads (Roche #11719416001) preblocked with salmon sperm (Thermo Fisher, # 15632011) were used to precipitate antibody complexes from diluted samples. Beads were sequentially washed at 4 °C with low-salt buffer (0.1% SDS, 1% Triton X-100, 2 mM EDTA, 20 mM Tris-HCl pH 8.0, 150 mM NaCl), high-salt buffer (0.1% SDS, 1% Triton X-100, 2 mM EDTA, 20 mM Tris-HCl pH 8.0, 500 mM NaCl), LiCl buffer (0.25 M LiCl, 1% NP-40, 1% sodium deoxycholate, 1 mM EDTA, 10 mM Tris-HCl pH 8.0), and TE buffer (10 mM Tris-HCl pH 8.0, 1 mM EDTA). Antibody complexes were eluted by incubating beads for 15 min at 30 °C in elution buffer (1% SDS, 100 mM $NaHCO_3$). Eluates were digested with proteinase K and RNAse A and DNA fragments were purified using a PCR purification kit (Macherey-Nagel, #740609).

## Immunofluorescence and microscopy

Primary antibodies used in this study are listed in the Key resources table. Note that the MANNES1E antibody detects different nesprin-1 isoforms (*Holt, 2016*; *Randles et al., 2010*). However, previous studies have shown that only the nesprin-1α isoform is upregulated during muscle differentiation (*Espigat-Georger et al., 2016*; *Gimpel et al., 2017*; *Holt, 2016*). Prior to fixation, cells were rinsed once with PBS. Cells were fixed either with pre-chilled methanol at –20 °C for 3 min or with 4% formaldehyde/PBS for 10 min at room temperature. Formaldehyde-fixed cells were permeabilized with 0.5% TritonX-100/PBS. Prior to antibody staining, samples were blocked for at least 20 min using 5% BSA in 0.2% Tween-20 in PBS. Primary antibodies were diluted in blocking reagent and incubated with the sample for 90 min at room temperature or overnight at 4 °C. After removal of primary antibody solution and three 5 min washes with 0.1% NP40/PBS, samples were incubated for 60 min with fluorophore-coupled secondary antibodies. DNA was visualized with 0.5 µg/ml DAPI (4′,6′-diamidino-2-phenylindole) in 0.1% NP40/PBS. After DAPI staining, cover slips were rinsed once with Millipore-filtered water and then mounted using Fluoromount-G mounting medium. Analysis, image acquisition, and high-resolution microscopy were done using a LSM800 confocal laser scanning microscope equipped with an Airyscan detector and the ZEISS Blue software (Carl Zeiss AG, RRID: SCR_013672) with Airyscan image processing.

## Image analysis

All image analyses were carried out using the Fiji software package (http://fiji.sc, RRID:SCR_002285). For quantification of nuclear envelope coverage, confocal images were transformed into binary images by setting a manual intensity threshold. Regions of interests (ROIs) were obtained by detecting nuclei outline via DAPI staining and subsequent transformation of these outlines into 1-µm-wide bands. Coverage was quantified as the percentage of positive pixels inside bands in the binary images. ROIs for measuring intensities at centrosomes were generated by detecting local signal maxima in γ-tubulin channels and subsequent generation of circular ROIs with 1 µm diameter using the maxima as centers.

To measure nuclear envelope intensity profiles for nesprin-1α and AKAP6, we first created ROIs by manually detected nuclear outlines using DAPI staining. We then decreased the diameter of these ROIs by 1 µm and used the newly created ROIS as starting points for linear intensity profiles perpendicular to the nuclear outlines.

## Microtubule regrowth assay

C2C12 cells were treated with 5 µM nocodazole (Sigma-Aldrich, Cat# M1404) in culture media for 3 hr at 37 °C to depolymerize microtubules. To observe microtubule regrowth, nocodazole-containing medium was removed, cells were rinsed three times with cold medium, and either fixed (0 min time

point) with 4% formaldehyde in PBS for 10 min or immediately transferred to 37 °C pre-warmed culture media for the desired length of time followed by formaldehyde fixation. Myotubes were extracted with 1% Triton X-100 in PHEM buffer (60 mM PIPES, 25 mM HEPES, 10 mM EGTA, 2 mM MgCl$_2$, pH 6.9) for 30 s at room temperature prior to fixation.

## Bioinformatics analysis

Data was analyzed with R (http://www.r-project.org/; RRID:SCR_001905) and Bioconductor (http://www.bioconductor.org/, RRID: SCR_006442). Myogenin ChIP-Seq data (GEO accession number: GSE36024) produced within the ENCODE project (*Consortium, 2012*; *Consortium et al., 2020*; *Yue et al., 2014*) were obtained via the UCSC Genome Browser at https://genome.ucsc.edu/index.html (*Kent et al., 2002*; *Rosenbloom et al., 2013*). NarrowPeak tracks relative to ChIP-seq data from four different time points were considered: undifferentiated C2C12 myoblasts as well as C2C12 cultures differentiated for 24 hr, 60 hr, or 7 days. To identify myogenin promoter binding, peaks were annotated to the Ensembl release 67 mouse genome relying on Bioconductor packages biomaRt v. 2.30.0 (*Durinck et al., 2009*) and ChIPpeakAnno v. 3.8.9 (*Zhu et al., 2010*). Genes were considered as myogenin targets if a peak (p-value $< 10^{-5}$) was localized at a maximum distance of 1 kb from the annotated transcriptional start site.

Results of differential expression analysis for RNA-seq data from C2C12 differentiation (GEO accession number: GSE84158) were obtained from the Gene Expression Omnibus (GEO) repository (*Doynova et al., 2017*). Three sample types were analyzed: C2C12 myoblasts (C1), C2C12 cultures differentiated for 3 days containing myoblasts as well as myotubes (C2), and C2C12 cultures differentiated for 7 days and treated with AraC, resulting in depletion of proliferating myoblasts (C3). Genes were considered as upregulated if they exhibited a positive fold change (p-value < 0.05) from C1 to C2 as well as from C1 to C3.

Gene expression microarray data for rat heart development were obtained as described previously using the Affymetrix GeneChip RAT 230 Expression Set (*Patra et al., 2011*). Genes were considered upregulated if (1) the corresponding probe set was identified as differentially expressed on the basis of a procedure that accounts for the total area under the profile compared to a constant profile (*Di Camillo et al., 2007*), and (2) the difference between the maximum expression value over time and the initial one was greater than the difference between the initial value and the minimum value.

For Gene Ontology analysis, annotated cellular component terms for each of the potential myogenin targets were retrieved using the search tool at http://geneontology.org/ (*Ashburner et al., 2000*; *The Gene Ontology Consortium, 2017*). Potential targets annotated with the terms 'nuclear membrane' and 'nuclear envelope' were considered.

## Quantification and statistical analysis

### Quantification of nesprin-1α+ nuclei

Low levels of nesprin-1 expression can be detected at nuclei of non-differentiated muscle cells using MANNES1E antibody. Prior to scoring of nesprin-1$\alpha$+ nuclei, we therefore set a threshold for nesprin-1 signal in images of differentiated C2C12 cells by measuring and subtracting maximal nesprin-1 signal intensity in undifferentiated C2C12 cultures.

### Statistical analysis

As preliminary experiments indicated a large effect size of siRNA treatments and ectopic MRF expression, three biological replicates were performed per experiment (i.e., n = 3). Biological replicate means that cells were freshly plated, treated, fixed and stained, and then analyzed. For each biological replicate, two technical replicates were performed in the sense that two individual wells were processed at the same time. When analyzed, the two technical replicates were scored as one sample.

For quantification in C2C12 cells, >500 nuclei were analyzed per condition and biological replicate. For MRF-GFP experiments, >50 GFP+ cells were analyzed per condition and biological replicate. For nuclear coverage and intensity quantifications of mScarlet and MYOG-mScarlet cells, >100 nuclei or centrosomes were analyzed per condition and pooled from three biological replicates to display distribution in violin plots.

Statistical analysis was carried out using GraphPad Prism 5.02 or Prism 8.2.1 (La Jolla, USA; RRID:SCR_002798). Differences between groups were considered statistically significant when the

p-value ≤ 0.05. The 95% confidence interval (CI) for the differences between compared groups are reported in the figure legends. Statistical significance of differences between groups was tested using the following:

*Figure 1F,H*, *Figure 1—figure supplement 1B,C*, *Figure 2—figure supplement 2A-C*, *Figure 4—figure supplement 1A,B*, *Figure 5B*, *Figure 5—figure supplement 3A,B*, *Figure 7E*: One-way ANOVA followed by Bonferroni's post hoc test to compare selected pairs of groups.

*Figure 2B*, *Figure 6C*, *Figure 7B,D,F*: Student's t-test together with an F-test to assess equality of variances.

*Figure 2D,F,H,M*, *Figure 3B,D,E*, *Figure 5D,E,F*, *Figure 5—figure supplement 2A,B*: Kolmogorov–Smirnov test to compare the cumulative distribution of groups.

## Acknowledgements

We acknowledge the Platform for Immortalization of Human Cells at the Institute de Myologie, Paris for the generation and distribution of immortalized human myoblasts. We thank Glenn E Morris for providing us with nesprin-1 antibody (MANNES1E). We thank the ENCODE consortium, the laboratory of Barbara Wold and the Millard and Muriel Jacobs Genetics and Genomics Laboratory at the California Institute of Technology for providing myogenin ChIP-Seq data. We thank Christina Warnecke for support with ChIP experiments, Marc Stemmler, Eva Bauer and Thomas Brabletz for their help with luciferase assays and Anna K Großkopf and Alexander S Hahn for experimental support. We acknowledge Les Laboratoires Servier for providing illustrations in the Servier Medical Art collection under a Creative Commons Attribution 3.0 Unported License (https://creativecommons.org/licenses/by/3.0/). We thank Anna K Großkopf, Manfred Frasch, Hanh Nguyen, Thomas U Mayer, Rosa M Puertollano, Payel Das, Marina Leone, Gentian Musa, and Salvador Cazorla-Vazquez for critical reading of the manuscript and all members of the Engel lab for critical discussions.

This work was supported by the Interdisciplinary Centre for Clinical Research Erlangen (IZKF project J42 to FF), the Emerging Fields Initiative Cell "Cycle in Disease and Regeneration" (CYDER to FBE) and an ELAN Program Grant (ELAN-16-01-04-1-Vergarajauregui to S.V.) from the Friedrich-Alexander-Universität Erlangen-Nürnberg, by the German Research Foundation (DFG, INST 410/91-1 FUGG and EN 453/12-1 to FBE), and by the Research Foundation Medicine at the University Clinic Erlangen, Germany.

## Additional information

### Funding

| Funder | Grant reference number | Author |
| --- | --- | --- |
| Interdisciplinary Center for Clinical Research (IZKF), Uniklinikum Erlangen | J42 | Fulvia Ferrazzi |
| Friedrich-Alexander-Universität Erlangen-Nürnberg | ELAN-16-01-04-1-Vergarajauregui | Silvia Vergarajauregui |
| Friedrich-Alexander-Universität Erlangen-Nürnberg | CYDER | Felix B Engel |
| Deutsche Forschungsgemeinschaft | INST 410/91-1 FUGG | Felix B Engel |
| Deutsche Forschungsgemeinschaft | EN 453/12-1 | Felix B Engel |
| Research Foundation Medicine at the University Clinic Erlangen | | Silvia Vergarajauregui Felix B Engel |

| Funder | Grant reference number | Author |
|---|---|---|

The funders had no role in study design, data collection and interpretation, or the decision to submit the work for publication.

## Author contributions

Robert Becker, Conceptualization, Investigation, Methodology, Visualization, Writing - original draft, Writing - review and editing; Silvia Vergarajauregui, Conceptualization, Funding acquisition, Investigation, Methodology, Supervision, Visualization, Writing - original draft, Writing - review and editing; Florian Billing, Conceptualization, Investigation, Methodology, Writing - review and editing; Maria Sharkova, Investigation; Eleonora Lippolis, Formal analysis, Investigation, Methodology; Kamel Mamchaoui, Formal analysis, Methodology, Resources; Fulvia Ferrazzi, Formal analysis, Funding acquisition, Methodology, Resources, Supervision, Writing - review and editing; Felix B Engel, Conceptualization, Formal analysis, Funding acquisition, Methodology, Supervision, Visualization, Writing - original draft, Writing - review and editing

## Author ORCIDs

Robert Becker (iD) http://orcid.org/0000-0001-7615-9390
Silvia Vergarajauregui (iD) http://orcid.org/0000-0002-9247-6123
Florian Billing (iD) http://orcid.org/0000-0002-3874-9012
Fulvia Ferrazzi (iD) http://orcid.org/0000-0003-4011-4638
Felix B Engel (iD) http://orcid.org/0000-0003-2605-3429

## Decision letter and Author response

Decision letter https://doi.org/10.7554/eLife.65672.sa1
Author response https://doi.org/10.7554/eLife.65672.sa2

---

# Additional files

## Supplementary files

- Supplementary file 1. Final list of myogenin targets.
- Supplementary file 2. List of oligonucleotides used for PCR and construct generation.
- Transparent reporting form

## Data availability

This work is based exclusively on the analysis of previously published data sets.

The following previously published datasets were used:

| Author(s) | Year | Dataset title | Dataset URL | Database and Identifier |
|---|---|---|---|---|
| Wold B, Jacobs M, Jacobs M, Marinov G, Fisher K, Kwan G, Kirilusha A, Mortazavi A, DeSalvo G, Williams B, Schaeffer L, Trout D, Antoschechkin I, Zhang L, Schroth G | 2012 | Transcription Factor Binding Sites by ChIP-seq from ENCODE/Caltech | https://www.ncbi.nlm.nih.gov/geo/query/acc.cgi?acc=GSE36024 | NCBI Gene Expression Omnibus, GSE36024 |
| O'Sullivan JM, Doynova MD, Cameron-Smith D, Markworth JF | 2017 | Transcriptome changes during the differentiation of myoblasts into myotubes | https://www.ncbi.nlm.nih.gov/geo/query/acc.cgi?acc=GSE84158 | NCBI Gene Expression Omnibus, GSE84158 |

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
