## [Decision Letter]

**Acceptance summary:**

The study identifies recruitment factors and their transcriptional regulation required for transferring MTOC activity from the centrosome to the nuclear envelope during differentiation of skeletal muscle cells. Understanding this process is important, since the non-centrosomal MTOC mediates the distribution of nuclei in multi-nucleated myotubes, which in turn is linked to muscle function and disease.

**Decision letter after peer review:**

[Editors’ note: the authors submitted for reconsideration following the decision after peer review. What follows is the decision letter after the first round of review.]

Thank you for submitting your work entitled "Myogenin controls via AKAP6 non-centrosomal microtubule organizing center formation at the nuclear envelope" for consideration by *eLife*. Your article has been reviewed by 3 peer reviewers, including Jens Lüders as the Reviewing Editor and Reviewer #1, and the evaluation has been overseen by a Senior Editor.

Our decision has been reached after consultation between the reviewers. Based on these discussions and the individual reviews below, we regret to inform you that your work will not be considered further for publication in *eLife*.

Overall, the manuscript presents interesting novel findings regarding the regulation of ncMTOC assembly at the nuclear envelope and most experiments appear well performed and presented. However, although the study identifies AKAP6 as a novel ncMTOC factor, it does not provide sufficient mechanistic insight. Is AKAP6 merely a bridge between Nesprin and AKAP450? What does it interact with, what other centrosome proteins are involved? How is its localization regulated to allow for the timely activation of the nucleus as an MTOC? Is AKAP6 sufficient to induce MTOC function? Considering these and other issues raised by all three reviewers, we feel that the manuscript does not reach the level of insight and impact that we seek for *eLife* at this stage. However, if the authors are able to address all of the raised issues, we would happily reconsider a new manuscript and review it in relation to these reviewer comments.

*Reviewer #1:*

The manuscript by Becker et al. provides novel insight into the regulation of the redistribution of microtubule organizing center (MTOC) function from centrioles to the nuclear envelope during muscle cell differentiation. Assembling the non-centrosomal MTOC on the surface of nuclei and inactivating the centrosome is linked to the different cytoplasmic organization of differentiated, multi-nucleated myotubes and is required for proper alignment of nuclei along the elongated myotubes. The demonstration that myogenin expression is sufficient to induce MTOC re-distribution and the identification of the crucial role of AKAP6 in assembling the nuclear envelope MTOC are important novel findings and advance our understanding of this process. In myotubes AKAP6 depleted nuclei do not align properly, as shown previously for myotubes lacking MTOC function at the nuclear envelope. Finally, while myogenin expression is not able to induce MTOC redistribution in cancer cells, this process can be observed upon fusion with myogenin-expressing fibroblasts. The authors claim that this may serve as strategy to attenuate centrosomal MTOC activity, which has been linked to cancer development and progression.

Overall, the study presents interesting novel findings, but lacks analytical depth regarding the underlying molecular mechanism. Also, several findings require more solid quantitative analysis. In particular, the part investigating attenuation of cancer centrosomes is quite weak in this regard.

1) I am surprised that the authors seem to observe only nuclei that are either positive or negative for MTOC markers. Is this process (redistribution of centrosome proteins) not gradual? I would suggest a different way of quantifying this process, to account for the temporal aspect and include intermediate stages.

2) Related to point 1: after myogenin expression, is the centrosomal MTOC inactivated first, after, or simultaneously with the appearance of the nuclear envelope MTOC? Are centrosomes in 2G active or inactive? Do 100% of cells look like the two examples in 2G?

3) The attenuation of the centrosomal MTOC should be quantified upon myogenin expression should be quantified. PCM1 is considered a centriolar satellite marker. The authors should use additionally centrosomal markers such as pericentrin and γ-tubulin to evaluate the state of the centrosomal MTOC.

4) 3D: the spatial distribution is not very clear from the image. A larger area of the nuclear envelope should be shown and some type of quantification (e.g. intensity profiles) that reveals the difference between nesprin and AKAP6. Are the locations of antibody epitopes (N-term. vs C-term.) consistent with the authors' interpretation?

5) 4C: a different centrosomal marker that is not affected by myogenin is needed, to identify the presence of centrosomes/centrioles in all cells. Also, PCM1 is a satellite marker. The disappearance of satellites may or may not indicate changes in centrosome staining. Moreover, as the assembly of the ncMTOC, the disassembly of the centrosomal MTOC needs to be quantified.

6) Figure 6: as before, other centrosome markers (including one that is unaffected) are needed to evaluate the state of the centrosomal MTOC. Higher magnifications are also required in particular of the centrosome region.

7) The experiments in figure 6 are not convincing for claiming that centrosome attenuation can be induced in cancer cells. First, the quality of the provided data is simply not enough. Is the amount of PCM reduced? Is the activity reduced? Second, after fusion they are technically not cancer cells anymore.

8) An important experiment would be to test whether nesprin-1alpha and AKAP6 ectopic expression can induce MTOC assembly at the nuclear envelope. If so, this may allow more detailed mechanistic studies (expression of mutants etc), to provide more depth to the study.

*Reviewer #2:*

Microtubule organization varies by cell type and function. Dividing cells, cancer cells, and some specialized cell types such as fibroblasts use centrosomes as microtubule organizing centers (MTOCs), but most differentiated cells employ non-centrosomal MTOCs (ncMTOCs) to obtain more complicated microtubule patterns during the process of differentiation. Despite decades worth of work on the centrosome, comparatively little is known about ncMTOC composition or how the cell regulates the switch in MTOC function from the centrosome to non-centrosomal sites during differentiation. Here Becker et al., use differentiated muscle cells as a model to understand how the well-defined nuclear ncMTOC, which is a hallmark of skeletal muscle, is established. Previous studies found that PCM proteins such as AKAP450, Pericentrin, and PCM1 localize to the nuclear envelope of muscle cells, that Nesprin1a recruits these proteins, and that Nesprin and AKA450 are required for microtubule nucleation at this site. Here Becker et al., appear to uncouple the expression of Nesprin and PCM; Nesprin and PCM1 are both expressed in cells positive for expression of the transcription factor MyoD, but Nesprin expression precedes expression of the transcription factor myogenin, and PCM1 but not Nesprin expression is controlled by myogenin. These results suggested that activation of MTOC function at the nucleus is downstream of Nesprin localizing there and requires expression of myogenin targets. Consistently, myogenin expression in naïve cells is sufficient to induce nuclear MTOC activation. By mining ChIP-seq, RNAseq and microarray data, the authors identify AKAP6 as this potential target, finding that AKAP6 localizes to the cytoplasmic face of the nucleus, fails to localize to the nucleus in SYNE1/Nesprin mutant patient derived myoblasts, and is required to localize PCM1, AKAP450, and Pericentrin to the nuclear envelope. AKAP6 also is required for microtubule regrowth from the nucleus of myotubes and for nuclear positioning in these cells, a MT controlled process. Finally, the authors find that fusion of a Myogenin induce muscle cell can convert a nive NIH3T# cells with a centrosome MTOC to a nuclear MTOC state. While I love a cell fusion experiment, this result is not surprising given that fusion of a myoblast to a U2OS cell can similar convert the non-muscle cell from a centrosome to a nuclear MTOC (Fant et al., 2009, not cited).

The experiments appear well preformed and presented. However, although these studies appear to identify a novel ncMTOC factor at the nucleus of muscle cells, they do little in the way of uncovering mechanistic aspects of how AKAP6 functions there or on assembly of the nuclear ncMTOC in these cells. Is AKAP6 merely a bridge between Nesprin and AKAP450? How is its localization regulated to allow for the timely activation of the nucleus as an MTOC? Is AKAP6 sufficient to induce MTOC function? In the absence of more mechanistic insight on the role of AKAP6, I am not sure that this paper will be of broad enough interest for the readership of *eLife*.

The findings here do little in the way of uncovering how AKAP6 functions at the nucleus in ncMTOC formation. This could be addressed in many ways including:

– Does AKAP6 bind to Nesprin and AKAP450 (or other MTOC components) to create a bridge? If so, how is this association regulated? Limited by expression of AKAP6?

– Is AKAP6 localization sufficient to promote MTOC activity? This could be addressed by artificially tethering AKAP6 to the plasma or mitochondrial membrane and seeing if MTOC proteins and microtubules are recruited. If this worked, the ability of such a structure to compete with the centrosome could also be directly addressed.

The authors repeatedly claim that 'Myogenin is required for centrosomal protein recruitment to the nuclear envelope', but they really only look at PCM1 localization in most cases. Why not look at AKAP450? This is especially important when arguing that Myogenein controls MTOC formation downstream of Nesprin localization to the nucleus.

If Nesprin expression is not controlled by myogenin, in naïve cells in which myogenin overexpressed, how does Nesprin become expressed and localized to the nucleus? The authors briefly comment on this, but I find their explanation unsatisfying especially given what they are arguing in Figure 1.

The fusion experiment is cool and suggests that the nuclear MTOC is dominant to a centrosomal MTOC in interphase cells. However, almost the exact experiment has been done but is not cited (Fant et al., 2009: "Nuclei of non-muscle cells bind centrosome proteins upon fusion with differentiating myoblasts") Similarly, although less directly relevant, dominance was tested between a mitotic centrosome MTOC and a membrane ncMTOC (Yang and Feldman, 2015, also not cited).

– Line 49: PCM needs to be defined as material that is specific to the centrosome. PCM proteins might localize to other locations, but 'pericentriolar' refers to the material surrounding centrioles.

– The authors interchange usage of expression and localization.

*Reviewer #3:*

The manuscript by Becker et al. focusses on the regulation of non-centrosomal microtubule organizing centers (ncMTOCs), specifically MTOC activity at the surface of the nucleus. The authors show that during muscle development, the MTOC activity transitions from the centrosome to the nucleus, and that this transition depends on the transcriptional activation of AKAP6 by myogenin, a TF that drives muscle differentiation. While it seems that several results can be found in the literature (such as the transition to nuclear ncMTOCs, Myogenin binding to AKAP6 promoter, AKAP6 binding to Nesprin), the authors assemble them together well in a single manuscript to better link muscle differentiation to ncMTOC formation. Overall I found the study very interesting and most of the conclusions/model supported by the data, although we seem to disagree on direct vs indirect links with APAP6. Here are my comments and suggestions.

A) Figure 1 is critical for setting up the order of events during differentiation. While the final conclusion of this section is correct, following the data in this section was extremely difficult. I had to read this section 4 times as I knew the data was in there somewhere, I just had to find it. I recommend replacing C and E with 4 graphs that show: 1) the % of the MyoD+ nuclei that are positive for PCM1, Myog, nesprin; 2) the % of PCM1+ nuclei that are positive for Myog, MoyD, nesprin; 3) the % of Myog+ nuclei that are positive for PCM1, MyoD, nesprin; 4) the % of nesprin+ nuclei that are positive for PCM1, MyoD, Myog. Then the description in the text should match these graphs.

B) Related to the use of PCM-1 as the main PCM marker and an indicator of MTOC activity. I always thought of PCM-1 as a centrosome satellite component, which has little direct function on centrosome MTOC activity. In fact, in the original paper describing PCM-1 (Balczon et al. 1994) it is shown to leave centrosomes in the lead up to, and throughout mitosis, when the centrosome is at its highest MTOC activity. PCM-1's role in ncMTOC activity might be better understood, but ultimately γ-tubulin (g-tub) must be recruited to the site of nucleation. Thus, it would be nice to see g-tub and MTs in several experiments such as 1D, 2A-F, 3E, 4F.

C) The model presented is that myognin binds AKAP6 promoter, increases expression, which drives AKAP6 binding to nesprin and triggers downstream PCM recruitment to the nuclear surface. While the order of events is nicely shown, a direct link upstream and downstream of AKAP6 is not. Many additional experiments must be performed to claim a direct role as shown in Figure 7. For example: mutating AKAP6 promoter, bypassing myog by overexpressing AKPA6, showing nuclear MTOCs when nesprin and AKAP6 are coexpressed in fibroblasts, bypassing nesprin by artificially tethering AKAP6 to nuclear envelope, mapping AKAP6 binding to PCM proteins and mutating that interaction site, and so on. If the reviewers and editor(s) feel this is beyond the scope of the manuscript, I suggest that the authors explicitly state, and show in the model, that indirect paths are possible/likely.

D) It is not clear, given the similarity in steady state MTs between control and si-AKAP6, why nuclear positioning is different. The manuscript would be strengthened if it ended with the role of ncMTOCs in nuclear spacing instead of the current figure 6, which I think is not necessary and should be removed. Can the authors follow MT nucleation and nuclear positioning during the development of myotubes in control and si-AKAP6? Are the MTs more dynamic or more stable with or without nuclear MTOCs? What are the difference between myotubes with aligned nuclei and overlapping nuclei in the si-AKAP6 populations? Are the ones that end up overlapping start out too close to one another and therefore require ncMTOC activity to push the nuclei apart? Can the authors provide any further insight here?

[Editors’ note: further revisions were suggested prior to acceptance, as described below.]

Thank you for submitting your article "Myogenin controls via AKAP6 non-centrosomal microtubule organizing center formation at the nuclear envelope" for consideration by *eLife*. Your article has been reviewed by 3 peer reviewers, including Jens Lüders as the Reviewing Editor and Reviewer #1, and the evaluation has been overseen by Suzanne Pfeffer as the Senior Editor.

Essential revisions:

1) The authors should clarify the relative regulation of MTOC activities. Centrosome inactivation may be due to factors that remove microtubule organizing capacity. Alternatively, the nuclear envelope MTOC could outcompete the centrosome by recruiting MTOC components. In cases where the nuclear envelope MTOC is perturbed, does the centrosome remain active? Does the centrosomal pool of PCM-1 remain prominent upon myogenin knockdown? After SYNE1 or AKAP6 knockdown, what is the state of the centrosome?

2) Since they are at the core of the regulation mechanism, the IP/CHIP experiments need extra controls. The authors should probe for a protein that is not specifically retained by the precipitating antibody.

3) What are the relative contributions of myoD versus myogenin in the regulation of MTOC assembly at the nuclear envelope? Experiments should be conducted to address: is myoD expression affected in the absence of myogenin? Does myoD expression also induce nesprin and AKAP6?

4) Please provide a zero time point in regrowth experiments and check nucleation activity in the experiment in Figure 7G. This is needed to argue that an MTOC is present.

5) The introduction and discussion should be improved by discussing:

– The relevance of the presented findings for muscle function and disease.

– The roles of the particular centrosomal proteins that are recruited to the nuclear envelope in generating and organizing microtubules.

– The term "centrosomal proteins". As they shuttle from centrosome to the nuclear envelope, the term "MTOC proteins" or similar may be more appropriate.

*Reviewer #1:*

The study "Myogenin controls via AKAP6 non-centrosomal microtubule organizing center formation at the nuclear envelope" by Becker et al. identifies the transcriptional regulation and the recruitment factors required for assembly of the MTOC at the nuclear envelope in skeletal muscle cells. Assembly of the non-centrosomal MTOC is still poorly understood, but is important for the distribution of nuclei in multi-nucleated myotubes, which in turn is linked to muscle function and disease. The authors show that the transcription factor myogenin drives this process and that AKAP6beta and nesprin-1alpha are crucial transcriptional targets. Overexpression of these two proteins is sufficient to drive nuclear envelope MTOC formation even in the absence of myogenin expression.

The manuscript extends previously published work by the same group on the role of AKAP6 in non-centrosomal MTOC assembly at the nuclear envelope. The current study elucidates the upstream factors that regulate this process and the provided data is of high quality.

The strength of this study is the identification of transcription factors and MTOC assembly factors that are sufficient to induce nuclear envelope MTOC assembly, including in non-muscle cells. This provides the first framework for a mechanistic understanding of this process and for further analysis in future studies.

Weaknesses are that experiments identifying transcriptional targets require additional controls and that the implications of the findings for muscle function and related diseases have not been described very well.

The authors have addressed my concerns raised during review of a previous version. Regarding this new manuscript, which has a different focus, there are some remaining issues that need to be addressed:

1) All IP/ChIP experiments lack controls (Figure 2, 4, 5). These are integral to the main claims of the paper and the authors should show control proteins that are detected in the input, but are not specifically retained by the precipitating antibody.

2) For the non-expert reader, the introduction and discussion need to be improved in terms of the relevance of the presented findings for muscle function and disease. What is the role of the nuclear envelope MTOC? How does it affect nuclei positioning, why is this important?

*Reviewer #2:*

Here, Becker et al. set out to identify the factors that control MTOC formation at the nuclear envelope of skeletal muscle cells. The nuclear envelope has long been appreciate for its ability to grow and localize microtubules, however the mechanisms that induce this state and simultaneously inactivate the centrosome as an MTOC have remained elusive. The authors identify the transcription factor myogenin as both necessary and sufficient to induce MTOC formation at the nuclear envelope. By mining the literature and ChipSeq and RNAseq data, they identify Nespin and AKAP6 as downstream targets of myogenin, that together are sufficient to induce a nuclear envelope MTOC in fibroblasts, a cell type that normally maintains microtubules at the centrosome and Golgi. Furthermore, myogenin is required to specifically induce cell-type specific isoforms, of these proteins. Loss of AKAP6 and results perturbs the recruitment of MTOC proteins to the nuclear envelope and leads to nuclear positioning defects in myotubes. Together, these data establish a module within the differentiation program of muscle cells for establishing the nuclear envelope as an MTOC. This is a resubmission of a former manuscript and in general, the authors have greatly expanded the work and added significant mechanistic insight.

1) I am left a bit confused about what the authors think are the relative contributions of myoD versus myogenin. It seems that the authors are setting up a linear pathway where MyoD induces expression of myogenin which in turn induces ncMTOC formation, especially since it is known that myoD activates myogenin transcription (Berkes et al., 2005). If this is indeed the pathway the authors are testing, they should explicitly state that at the beginning of this section rather than at the end. Additionally, appropriate experiments should be conducted, i.e. is myoD expression affected in the absence of myogenin?; does myoD expression also induce nesprin and AKAP6 expression?

2) The switch in MTOC function from the centrosome to non-centrosomal sites in differentiated cell is a general and interesting phenomenon. The authors propose that centrosome inactivation must be due to the expression of factors that remove microtubule organizing capacity from the centrosome. Alternatively, the nuclear envelope MTOC activity could outcompete the centrosome, stealing their shared components. Given that the authors have numerous cases where the nuclear envelope MTOC is perturbed, can they comment on whether the centrosome remains active? For example, what appears to be the centrosomal pool of PCM-1 remains prominent upon myogenin KD. Is this the case? Similarly, in the SYNE1 or AKAP6 knock down, what is the state of the centrosome?

3) Regrowth experiments need a zero timepoint to be able to assess how well microtubules were removed upon depolymerization treatments. As presented, these experiments are hard to interpret. While it is unquestionable that microtubules are associated with the nuclear envelope in these cases, it is important to distinguish whether these are new microtubules or persistent anchored, stabilized ones.

4) Does nesprin directly bind to AKAP6? The authors indicate that the interaction is direct.

5) Figure 7G is very nice, but it would be much nicer to see that microtubules also grow from the nuclear envelope in this case in addition to the fact that PCM-1 is recruited.

6) I do not think the problem and players are set up well enough for the broad readership of *eLife*. The introduction and/or discussion could be more extensive. For example, there is no discussion of why recruitment of these particular centrosomal proteins to the nuclear envelope would be important to generate microtubules. I also think it is important to explain what "centrosomal proteins" are as they clearly do not stay at the centrosome and instead shuttle to the nuclear envelope. Perhaps the authors could use "MTOC proteins" or something similar.

*Reviewer #3:*

The manuscript by Becker et al. is a new submission of previously *eLife* submission that focusses on the transition from centrosomal MTOCs to non-centrosomal MTOCs, specifically the surface of the nucleus. This submission more clearly focuses on triggering ncMTOC transition in developing skeletal muscles via the function of MyoD and Myogenin, members of the transcription factors family termed myogenic regulatory factors (MRFs). The final model shown in Figure 8 is well supported by the presented data, which I believe is much improved over the previous manuscript. Reading the recently published work from the same group (Vergarajauregui 2020), which appears to have been in review at the same time as the original submission of this manuscript, helped clarify some of the interaction and functional claims downstream of AKAP6. The authors have done a great job addressing all my previous concerns, they should be commended for this and my comments below are mostly small changes that do not move the goal line for the authors.

1) In several cases in the beginning of the manuscript (ex. First paragraph pf the results) it was not clear that ncMTOC specifically referred to nuclear envelope MTOCs. I would recommend explicitly stating in this first result section that throughout the manuscript, ncMTOCs refers to the nuclear envelope in this study. Alternatively, maybe the authors can use NE-MTOC to distinguish it from other ncMTOCs.

2) On line ~181 the authors state that "both MRFs are sufficient individually to induce a switch from centrosomal MTOC to a nuclear envelope MTOC." At this point in the manuscript the authors had not yet shown the elimination of centrosomes, that comes in figure 3. I suggest removing a reference to a switch at this point.

3) In several location in the manuscript the author state that myogenin is required to recruit centrosomal proteins. This can be found in lines 193 and 570 and other places. It is more accurate to say that myogenin is require for the localization of centrosome proteins to the nuclear envelope. Recruitment implies myogenin is the recruiting protein, which it is not, it is simply upstream in the ncMTOC formation pathway.

4) The figures overall are much improved and the quantification is suitable. However, It is important to report the Ns in each of the experiments. For example in figure 3, how many nuclei were measured? How many experimental repeats were performed.

5) Presentation of the analysis leading up to AKAP6 was great

6) How reliable is the specificity of the E-Box PCR amplification? Have the authors confirmed specificity of the ChIP experiment by PCR amplifying other regions of the AKAP and nesprin promotors? Although, it does appear to be specific enough to distinguish the different isoforms.

7) Direct protein interaction cannot be determined by IPs. The authors should change the conclusion that "AKAP6 localizes to the nuclear envelope in differentiated skeletal muscle cells via interaction with nesprin-1α" to AKAP6 localization depends on nesprin-1a. An interaction can only be determine using purified proteins or Y2H as the authors did with AKAP6 and PACT in the (Vergarajauregui 2020). If there is direct protein interaction data, the authors should present it, otherwise the current results should be presented in terms of being upstream or downstream.

Another example of this is "AKAP6 is required for the localization of centrosomal proteins to the nuclear envelope, in part by interacting with PCM-1." If there is no direct protein interaction data available (here or in the literature), then the word interacting should be replaced by recruiting.

[Editors’ note: further revisions were suggested prior to acceptance, as described below.]

Thank you for submitting your article "Myogenin controls via AKAP6 non-centrosomal microtubule organizing center formation at the nuclear envelope" for consideration by *eLife*. Your article has been reviewed by a Reviewing Editor and Suzanne Pfeffer as the Senior Editor.

The Reviewing Editor has drafted this to help you prepare a revised submission.

Suggested revisions:

The paper is very much improved and you have addressed all the remaining points raised by the reviewers. I only have a couple of suggestions before formal acceptance.

1) It seems odd to not mention your recent work on AKAP6 published in *eLife* in the introduction and the result, where you describe identification of AKAP6. I would include statements referring to this work.

2) You showed that recombinant nesprin and AKAP6 expression is sufficient to recruit MTOC protein to the NE, but your new data now show that this is not sufficient to generate an active MTOC. In my opinion this finding is very important and deserves a discussion. It suggests that myogenin induces another factor that is crucial for MTOC activity. While you correctly state "recruitment of MTOC proteins" rather than "MTOC formation" in the paper, non-expert readers may assume that this is synonymous. Without clarification/discussion this could be misleading.

---

## [Author Response]

Overall, the manuscript presents interesting novel findings regarding the regulation of ncMTOC assembly at the nuclear envelope and most experiments appear well performed and presented. However, although the study identifies AKAP6 as a novel ncMTOC factor, it does not provide sufficient mechanistic insight. Is AKAP6 merely a bridge between Nesprin and AKAP450? What does it interact with, what other centrosome proteins are involved? How is its localization regulated to allow for the timely activation of the nucleus as an MTOC? Is AKAP6 sufficient to induce MTOC function? Considering these and other issues raised by all three reviewers, we feel that the manuscript does not reach the level of insight and impact that we seek for eLife at this stage. However, if the authors are able to address all of the raised issues, we would happily reconsider a new manuscript and review it in relation to these reviewer comments.

As discussed with the editor, several of the raised issues have been addressed in the companion paper, which has recently been accepted for publication.

While the companion paper focussed on the elucidation of the role of AKAP6 as a critical player for ncMTOC formation in cardiomyocytes and osteoclasts, the focus of this manuscript is to elucidate the initiation of ncMTOC formation in vertebrate cells.

In summary, the present manuscript together with the companion paper identify AKAP6 as a critical player for ncMTOC formation in the three vertebrate cell types known so far to have a nuclear envelope MTOC: skeletal muscle cells, cardiomyocytes, and osteoclasts. We illuminate the upstream regulation of ncMTOC formation, showing that a single transcription factor can induce the attenuation of the centrosomal MTOC and formation of the ncMTOC at the nuclear envelope. We further give mechanistic insight into how AKAP6 functions as a critical adaptor between nesprin-1 and centrosomal proteins to create a ncMTOC and to localize the Golgi in close vicinity to the nuclear envelope. Furthermore, both manuscripts show the functional significance of AKAP6 for microtubule- or Golgi-dependent processes in skeletal muscle cells, cardiomyocytes, and osteoclasts. Due to the different experimental approaches and model systems as well as the large variety of addressed points/questions, we strongly believe that both manuscripts deserve publication in *eLife*.

Reviewer #1:[…] Overall, the study presents interesting novel findings, but lacks analytical depth regarding the underlying molecular mechanism. Also, several findings require more solid quantitative analysis. In particular, the part investigating attenuation of cancer centrosomes is quite weak in this regard.1) I am surprised that the authors seem to observe only nuclei that are either positive or negative for MTOC markers. Is this process (redistribution of centrosome proteins) not gradual? I would suggest a different way of quantifying this process, to account for the temporal aspect and include intermediate stages.

We thank the reviewer for indicating this point. In our C2C12 experiments, <2% of total PCM1-positive nuclei showed clear intermediate stages and have therefore been included in the total percentage of PCM-1-positive nuclei. To clarify this issue, we included this information in the text and provide a Supplementary Figure with intermediate stages of PCM-1 redistribution (Figure 1 – Supplemntal Figure 1A). For pericentrin and AKAP9 in C2C12 we have manually distinguished between partially and fully positive nuclei, as an automated analysis was not possible due to high cell density required for C2C12 differentiation. However, an automated analysis was performed for ectopic myogenin expression experiments in NIH3T3 cells, which could be grown at lower density. In this case nuclear envelope coverage by centrosomal proteins was quantified to account for the temporal aspect of recruitment (Figure 2).

2) Related to point 1: after myogenin expression, is the centrosomal MTOC inactivated first, after, or simultaneously with the appearance of the nuclear envelope MTOC?

Our new data provided in Figure 2 and 3 indicate that attenuation of the centrosomal MTOC and activation of the ncMTOC occur simultaneously. In order to clarify this issue, we created a cell line with tetracyclineinducible myogenin, which yields a high number of cells that form an ncMTOC and therefore allows reliable quantification. We quantified ncMTOC formation (Figure 2) and centrosome attenuation (Figure 3) upon myogenin expression. Furthermore, microtubule regrowth experiments showed that, while centrosomes are attenuated, microtubules still regrow from centrioles alongside growth from the nuclear envelope (Figure 3 F). In addition, -tubulin signal intensity at centrioles was quantified.

Are centrosomes in 2G active or inactive?

As described under point 2a, our analysis indicates that centrosomes are attenuated upon myogenin expression but retain some microtubule nucleating activity.

Do 100% of cells look like the two examples in 2G?

To address this issue, we have quantified the nuclear coverage by centrosomal proteins as well as their intensity at the centrosome upon myogenin expression and displayed the results as violin plots to visualize frequency distributions in Figure 2 and Figure 3.

3) The attenuation of the centrosomal MTOC should be quantified upon myogenin expression should be quantified. PCM1 is considered a centriolar satellite marker. The authors should use additionally centrosomal markers such as pericentrin and γ-tubulin to evaluate the state of the centrosomal MTOC.

We thank the reviewer for these suggestions and have now quantified centrosome attenuation in Figure 3 based on pericentrin and -tubulin.

4) 3D: the spatial distribution is not very clear from the image. A larger area of the nuclear envelope should be shown and some type of quantification (e.g. intensity profiles) that reveals the difference between nesprin and AKAP6. Are the locations of antibody epitopes (N-term. vs C-term.) consistent with the authors' interpretation?

As requested, we provide quantifications and better images of a larger area of the nuclear envelope (Figure 4 and Figure 4—figure supplement 1). The antibody epitopes do not allow a conclusion in regards to our interpretation. The nesprin-1 antibody was raised against the full-length nesprin-1 and a few amino acids of nesprin-1 (Randles et al., 2010, Dev Dyn). However, its epitope could not be mapped so far (Randles et al., 2010 Dev Dyn).

Locations of the antibody epitopes:

AKAP6 (HPA048741 Σ, rb polyclonal): 752-891 of 2319 aa total; 1^st^ spectrin domain nesprin-1 (monoclonal raised against full-length nesprin-1 plus an Nterminal extension into the nesprin1 sequence by 121 aa): epitope has not been mapped

5) 4C: a different centrosomal marker that is not affected by myogenin is needed, to identify the presence of centrosomes/centrioles in all cells. Also, PCM1 is a satellite marker. The disappearance of satellites may or may not indicate changes in centrosome staining. Moreover, as the assembly of the ncMTOC, the disassembly of the centrosomal MTOC needs to be quantified.

To address this issue, we now included -tubulin in all figures regarding centrosome activity. While -tubulin levels at centrosomes were affected by myogenin expression, the signal-to-noise ratio of -tubulin was still better than those of centriole markers like Cep135 and Centrin in our system and therefore allowed a reliable identification of centrosomes.

6) Figure 6: as before, other centrosome markers (including one that is unaffected) are needed to evaluate the state of the centrosomal MTOC. Higher magnifications are also required in particular of the centrosome region.7) The experiments in figure 6 are not convincing for claiming that centrosome attenuation can be induced in cancer cells. First, the quality of the provided data is simply not enough. Is the amount of PCM reduced? Is the activity reduced? Second, after fusion they are technically not cancer cells anymore.

Due to the extended critique on Figure 6 from all three reviewers, we will omit the current Figure 6. Instead we focused on providing more data on the initiation of ncMTOC formation.

8) An important experiment would be to test whether nesprin-1alpha and AKAP6 ectopic expression can induce MTOC assembly at the nuclear envelope. If so, this may allow more detailed mechanistic studies (expression of mutants etc), to provide more depth to the study.

Previously, it has been reported that overexpression of nesprin-1 is sufficient in myoblasts to recruit the PACT domain and small amounts of endogenous PCM-1 (Espigat-Georger et al., 2016 J Cell Sci; Gimpel et al., 2017 Curr Biol). However, this recruitment was inefficient.

To address the raised issue, we ectopically expressed nesprin-1 and AKAP6 in undifferentiated C2C12 myoblast and observed that nesprin-1 and AKAP6 are sufficient to recruit endogenous PCM-1 to the nuclear envelope. A more detailed study of AKAP6 domains and mutants is part of our recently accepted manuscript. The focus of the present manuscript was the upstream regulation of ncMTOC induction, where we show that myogenin preferentially activates the promoters of nesprin-1 and AKAP6 which in turn recruit centrosomal proteins and enable microtubule nucleation from the nuclear envelope.

Reviewer #2:Microtubule organization varies by cell type and function. Dividing cells, cancer cells, and some specialized cell types such as fibroblasts use centrosomes as microtubule organizing centers (MTOCs), but most differentiated cells employ non-centrosomal MTOCs (ncMTOCs) to obtain more complicated microtubule patterns during the process of differentiation. Despite decades worth of work on the centrosome, comparatively little is known about ncMTOC composition or how the cell regulates the switch in MTOC function from the centrosome to non-centrosomal sites during differentiation. Here Becker et al., use differentiated muscle cells as a model to understand how the well-defined nuclear ncMTOC, which is a hallmark of skeletal muscle, is established. Previous studies found that PCM proteins such as AKAP450, Pericentrin, and PCM1 localize to the nuclear envelope of muscle cells, that Nesprin1a recruits these proteins, and that Nesprin and AKA450 are required for microtubule nucleation at this site. Here Becker et al., appear to uncouple the expression of Nesprin and PCM; Nesprin and PCM1 are both expressed in cells positive for expression of the transcription factor MyoD, but Nesprin expression precedes expression of the transcription factor myogenin, and PCM1 but not Nesprin expression is controlled by myogenin. These results suggested that activation of MTOC function at the nucleus is downstream of Nesprin localizing there and requires expression of myogenin targets. Consistently, myogenin expression in naïve cells is sufficient to induce nuclear MTOC activation. By mining ChIP-seq, RNAseq and microarray data, the authors identify AKAP6 as this potential target, finding that AKAP6 localizes to the cytoplasmic face of the nucleus, fails to localize to the nucleus in SYNE1/Nesprin mutant patient derived myoblasts, and is required to localize PCM1, AKAP450, and Pericentrin to the nuclear envelope. AKAP6 also is required for microtubule regrowth from the nucleus of myotubes and for nuclear positioning in these cells, a MT controlled process. Finally, the authors find that fusion of a Myogenin induce muscle cell can convert a nive NIH3T# cells with a centrosome MTOC to a nuclear MTOC state. While I love a cell fusion experiment, this result is not surprising given that fusion of a myoblast to a U2OS cell can similar convert the non-muscle cell from a centrosome to a nuclear MTOC (Fant et al., 2009, not cited).The experiments appear well preformed and presented. However, although these studies appear to identify a novel ncMTOC factor at the nucleus of muscle cells, they do little in the way of uncovering mechanistic aspects of how AKAP6 functions there or on assembly of the nuclear ncMTOC in these cells. Is AKAP6 merely a bridge between Nesprin and AKAP450? How is its localization regulated to allow for the timely activation of the nucleus as an MTOC? Is AKAP6 sufficient to induce MTOC function? In the absence of more mechanistic insight on the role of AKAP6, I am not sure that this paper will be of broad enough interest for the readership of eLife.

We thank the reviewer for this feedback. Please note, that mechanistic details on the role of AKAP6 at the ncMTOC has been elucidated in the companion paper, which has just been published. The focus of this manuscript is to elucidate the initiation of ncMTOC formation in vertebrate cells, as mechanisms that initiate the switch from centrosomal to non-centrosomal MTOCs during differentiation of vertebrate cells are unknown. The only mechanistic insight into ncMTOC induction has been gained by studying *Drosophila* identifying the transcription factor trachealess to be required for ncMTOC formation (Brodu et al., 2010). The major finding of our study is the identification of the first transcription factor sufficient for induction of an ncMTOC and centrosomal MTOC attenuation.

The findings here do little in the way of uncovering how AKAP6 functions at the nucleus in ncMTOC formation. This could be addressed in many ways including:– Does AKAP6 bind to Nesprin and AKAP450 (or other MTOC components) to create a bridge?

This is not the scope of this manuscript and has been addressed in detail in the published companion manuscript.

If so, how is this association regulated? Limited by expression of AKAP6?

In this manuscript, we demonstrate by several experiments that AKAP6 expression is required for ncMTOC formation. In addition, we show that myogenin directly regulates AKAP6 expression. Finally, we also demonstrate that overexpression of AKAP6 and nesprin-1 is sufficient to recuirt endogenous centrosomal proteins.

– Is AKAP6 localization sufficient to promote MTOC activity? This could be addressed by artificially tethering AKAP6 to the plasma or mitochondrial membrane and seeing if MTOC proteins and microtubules are recruited. If this worked, the ability of such a structure to compete with the centrosome could also be directly addressed.

This is not the scope of this manuscript and has been addressed in detail in the published companion manuscript. Here, we show that overexpression of AKAP6 and nesprin-1 is sufficient to recuirt endogenous centrosomal proteins. Yet, the more important finding is that myogenin expression in fibroblasts was sufficient to initiate the formation of a functional ncMTOC.

The authors repeatedly claim that 'Myogenin is required for centrosomal protein recruitment to the nuclear envelope', but they really only look at PCM1 localization in most cases. Why not look at AKAP450? This is especially important when arguing that Myogenein controls MTOC formation downstream of Nesprin localization to the nucleus.

We thank the reviewer for this suggestion. In order to substantiate our conclusions, we expanded our analysis by quantifying pericentrin, AKAP9, and/or microtubule growth in myogenin-depleted C2C12 cells as well as myogenin overexpression cells.

If Nesprin expression is not controlled by myogenin, in naïve cells in which myogenin overexpressed, how does Nesprin become expressed and localized to the nucleus? The authors briefly comment on this, but I find their explanation unsatisfying especially given what they are arguing in Figure 1.

To clarify this issue, we performed chromatin immunoprecipitation and luciferase experiments, which demonstrate that myogenin binds and activates the nesprin-1 promoter in NIH3T3 fibroblasts.

The fusion experiment is cool and suggests that the nuclear MTOC is dominant to a centrosomal MTOC in interphase cells. However, almost the exact experiment has been done but is not cited (Fant et al., 2009: "Nuclei of non-muscle cells bind centrosome proteins upon fusion with differentiating myoblasts") Similarly, although less directly relevant, dominance was tested between a mitotic centrosome MTOC and a membrane ncMTOC (Yang and Feldman, 2015, also not cited).

We thank the reviewer for pointing out the missing references to existing literature and apologize for it. Due to the extended critique on Figure 6 from all three reviewers, we will omit the current Figure 6. Instead we focused on providing more data on the initiation of ncMTOC formation.

– Line 49: PCM needs to be defined as material that is specific to the centrosome. PCM proteins might localize to other locations, but 'pericentriolar' refers to the material surrounding centrioles.

The text will be changed as requested.

– The authors interchange usage of expression and localization.

The text will be adjusted.

Reviewer #3:The manuscript by Becker et al. focusses on the regulation of non-centrosomal microtubule organizing centers (ncMTOCs), specifically MTOC activity at the surface of the nucleus. The authors show that during muscle development, the MTOC activity transitions from the centrosome to the nucleus, and that this transition depends on the transcriptional activation of AKAP6 by myogenin, a TF that drives muscle differentiation. While it seems that several results can be found in the literature (such as the transition to nuclear ncMTOCs, Myogenin binding to AKAP6 promoter, AKAP6 binding to Nesprin), the authors assemble them together well in a single manuscript to better link muscle differentiation to ncMTOC formation. Overall I found the study very interesting and most of the conclusions/model supported by the data, although we seem to disagree on direct vs indirect links with APAP6. Here are my comments and suggestions.A) Figure 1 is critical for setting up the order of events during differentiation. While the final conclusion of this section is correct, following the data in this section was extremely difficult. I had to read this section 4 times as I knew the data was in there somewhere, I just had to find it. I recommend replacing C and E with 4 graphs that show: 1) the % of the MyoD+ nuclei that are positive for PCM1, Myog, nesprin; 2) the % of PCM1+ nuclei that are positive for Myog, MoyD, nesprin; 3) the % of Myog+ nuclei that are positive for PCM1, MyoD, nesprin; 4) the % of nesprin+ nuclei that are positive for PCM1, MyoD, Myog. Then the description in the text should match these graphs.

We changed Figure 1 and the corresponding section in the manuscript to clarify this issue. While we did not include all four suggested graphs, the new Figure 1 shows the percentage of C2C12 nuclei positive for MyoD, myogenin, nesprin-1, and PCM-1. In addition, the percentage of nesprin1-positive nuclei that are positive for MyoD and myogenin is shown and the text has been changed to clarify that always 100% of PCM-1-positive nuclei were postivie for MyoD, myogenin, and nesprin-1.

B) Related to the use of PCM-1 as the main PCM marker and an indicator of MTOC activity. I always thought of PCM-1 as a centrosome satellite component, which has little direct function on centrosome MTOC activity. In fact, in the original paper describing PCM-1 (Balczon et al. 1994) it is shown to leave centrosomes in the lead up to, and throughout mitosis, when the centrosome is at its highest MTOC activity. PCM-1's role in ncMTOC activity might be better understood, but ultimately γ-tubulin (g-tub) must be recruited to the site of nucleation. Thus, it would be nice to see g-tub and MTs in several experiments such as 1D, 2A-F, 3E, 4F.

The reviewer is right in the point that the amount of PCM-1 at the centrosome also changes during the normal cell cycle. The analysis of PCM-1 was chosen, as it is an integral part of centriolar satellites that play an important role in the transport of centrosomal proteins and centrosome assembly (Prosser and Pelletier, 2020 J Cell Sci). For example, it has been shown that PCM-1 depletion impairs recruitment of ninein and Pcnt to the centrosome and disturbs microtubule organization (Dammermann and Merdes, 2002 J Cell Biol).

To address this issue, we included the information regarding PCM-1 in the text and expanded our analysis by analyzing microtubule regrowth as well as the centrosomal proteins pericentrin and AKAP9, the latter of which has been shown to be essential for microtubule nucleation in muscle cells (Espigat-Georger et al., 2016 J Cell Sci; Gimpel et al. 2017 Curr Biol).

C) The model presented is that myognin binds AKAP6 promoter, increases expression, which drives AKAP6 binding to nesprin and triggers downstream PCM recruitment to the nuclear surface. While the order of events is nicely shown, a direct link upstream and downstream of AKAP6 is not. Many additional experiments must be performed to claim a direct role as shown in Figure 7. For example: mutating AKAP6 promoter/

Using chromatin immunoprecipitation and luciferase assays we added data to show that myogenin preferentially binds and activates the promoter of the muscle-specific -isoform of AKAP6.

Bypassing myog by overexpressing AKPA6, showing nuclear MTOCs when nesprin and AKAP6 are coexpressed in fibroblasts.

To address the raised issue, we ectopically expressed nesprin-1 and AKAP6 in undifferentiated C2C12 myoblast and observed that nesprin-1 and AKAP6 are sufficient to recruit endogenous PCM-1 to the nuclear envelope. Additionally, we show that PCM-1 co-immunoprecipiates with AKAP6. Together with the finding that depletion of AKAP6 results in a loss of centrosomal proteins from the nuclear envelope while nesprin-1 remains unaffected, this indicates that AKAP6 acts as an adapter between the nuclear envelope anchor nesprin-1 and centrosomal proteins.

Bypassing nesprin by artificially tethering AKAP6 to nuclear envelope, mapping AKAP6 binding to PCM proteins and mutating that interaction site, and so on.

This issues are beyond the scope of this manuscript and have been addressed in the published companion manuscript.

If the reviewers and editor(s) feel this is beyond the scope of the manuscript, I suggest that the authors explicitly state, and show in the model, that indirect paths are possible/likely.

We adjusted Figure 8 (former Figure 7) to fit our new findings.

D) It is not clear, given the similarity in steady state MTs between control and si-AKAP6, why nuclear positioning is different. The manuscript would be strengthened if it ended with the role of ncMTOCs in nuclear spacing instead of the current figure 6, which I think is not necessary and should be removed.

As suggested, we removed the former Figure 6.

The focus of the manuscript is the initiation of ncMTOC. Therefore, the manuscript end now on demonstrating that myogenin binds and activates the promoters of AKAP6 and nesprin-1a and that overexpresiion of AKAP6 and nesprin-1a is sufficient to induce the recruitment of endogenous centrosomal proteins to the nuclear envelope. Nuclear positioning was mainly utilized to validate the importance of AKAP6 for ncMTOC function. In order to address the raised questions regarding nuclear positioning, live cell imaging needs to be established which is beyond the scope of this study. Yet, we have added several data to provide further insides into the role of AKAP6 in nuclear positioning (see answers below).

Can the authors follow MT nucleation and nuclear positioning during the development of myotubes in control and si-AKAP6?

To understand the role of AKAP6 in nuclear positioning, we have depleted AKAP6 in myotubes and performed microtubule regrowth assay. Our data show, that AKAP6 depletion abolished microtubule growth from the nuclear envelope and resulted consequently in defects in nuclear positioning.

Are the MTs more dynamic or more stable with or without nuclear MTOCs?

To address this issue we have assessed the intensity of detyronisated microtubules a marker for stable microtubules. Our data reveal that AKAP depletion resulted in a reduction of detyronisated microtubules.

What are the difference between myotubes with aligned nuclei and overlapping nuclei in the si-AKAP6 populations? Are the ones that end up overlapping start out too close to one another and therefore require ncMTOC activity to push the nuclei apart? Can the authors provide any further insight here?

Early nucleus positioning in myotubes after fusion can be divided in two sub-processes. First, nuclei that cluster in the center of the newly formed myotube are pushed apart to spread along the length of the myotube.

This process is dependent on AKAP9-mediated microtubule nucleation (Gimpel et al., 2017 Curr Biol). Second, nuclei are aligned on the long axis of the cell. This process is mediated by microtubule-associated motors that are at least in part recruited to the nuclear envelope via PCM1 (Espigat-Georger et al., 2016 J Cell Sci). AKAP6 is potentially required for both processes by recruiting AKAP9 as well as PCM1 to the nuclear envelope. We quantified defects in nuclei spreading (i.e. overlapping nuclei) and nuclei alignment separately to assess the effect of AKAP6 depletion on each process.

To clarify this issue, we included a more detailed description in the text and a model of how AKAP6 depletion affects nuclear positioning in Figure 6. In addition, we confirmed that AKAP6 depletion impaired nuclear envelope localization of the dynein activator p150glued to the nuclear envelope in myotubes.

[Editors’ note: further revisions were suggested prior to acceptance, as described below.]

Essential revisions:1) The authors should clarify the relative regulation of MTOC activities. Centrosome inactivation may be due to factors that remove microtubule organizing capacity. Alternatively, the nuclear envelope MTOC could outcompete the centrosome by recruiting MTOC components. In cases where the nuclear envelope MTOC is perturbed, does the centrosome remain active?

Thank you very much for raising this important point. As detailed below, we have performed the suggested experiments to clarify the regulation of MTOC activity at the nuclear envelope and the centrosome and added a paragraph to the discussion (starting at line 553).

Does the centrosomal pool of PCM-1 remain prominent upon myogenin knockdown?

To address this issue, we have analyzed PCM-1 staining patterns in myogenin-depleted differentiating C2C12. The results are now described in lines 146-155 and representative images are shown in Figure 1—figure supplement 2. Importantly, in myogenin-depleted cultures, PCM-1 remains localized in a centrosomal pattern in cells with highly nesprin-1-positive nuclei. This suggests that endogenous myogenin contributes to the attenuation of the centrosome during NE-MTOC formation, which is in agreement with the observation that ectopic myogenin expression in fibroblast is sufficient to attenuate the centrosome.

After SYNE1 or AKAP6 knockdown, what is the state of the centrosome?

In order to clarify whether myogenin induces loss of MTOC proteins from the centrosome by establishing an NE-MTOC that outcompetes the centrosome or induces separate factors to attenuate the centrosome, we have depleted nesprin-1 or Akap6 in MYOG-mScarlet cells and analyzed pericentrin as well as γ-tubulin intensities at the centrosome. The analysis is now described in lines 393-399 and shown in Figure 5—figure supplement 2, revealing that levels of MTOC proteins at the centrosome are still reduced upon MYOG expression in nesprin-1- or AKAP6-depleted cells, meaning disturbing of the NE-MTOC does not prevent centrosome attenuation. These results indicate that competition between centrosome and NE-MTOC is not the mechanism underlying the attenuation of the centrosome during NE-MTOC formation.

2) Since they are at the core of the regulation mechanism, the IP/CHIP experiments need extra controls. The authors should probe for a protein that is not specifically retained by the precipitating antibody.

We are not sure what experiment the reviewer suggests. We assume that the reviewer would like additional controls regarding the specificity of the anti-myogenin antibody or other data that substantiate our conclusion that myogenin binds to the promoter regions of *Syne1* or *Akap6*. To address this issue, we have included an intronic region of *Syne1* or *Akap6* as negative control as well as a promoter region of *Desmin* as positive control for the specificity of the myogenin antibody in the ChIP experiments. Additionally, we have probed lysates of Tet-ON mScarlet cells as well as non-induced Tet-ON MYOG-mScarlet cells. The results of these new experiments, presented in Figure 2 – supplemental figure 2 and Figure 4 – supplemental figure 1 and described in lines 242-251 and lines 345-349, further show that myogenin binds the promoter regions of *Syne1* and *Akap6*.

3) What are the relative contributions of myoD versus myogenin in the regulation of MTOC assembly at the nuclear envelope? Experiments should be conducted to address: is myoD expression affected in the absence of myogenin?

To clarify this point, we have analyzed MyoD expression in myogenin-depleted C2C12 cells using RT-PCR and found that myogenin depletion does not affect *Myod* RNA levels. As positive control, we show that depletion of MyoD also reduces myogenin levels, which is consistent with literature as well as our findings that MyoD induces myogenin expression in fibroblasts. These results have been added as panel C to Figure 2 – supplemental figure 1 and described in lines 196-202.

Does myoD expression also induce nesprin and AKAP6?

Analysis of MyoD-GFP-transfected NIH3T3 fibroblasts by immunostaining revealed that nesprin-1α and AKAP6 expression is induced (Figure 5 – supplemental figure 3). This is in accordance with our previous finding that overexpression of MyoD in this cell line induces myogenin expression as well as PCM-1 recruitment to the nuclear envelope (Figure 2 – supplemental figure 1). In order to address the relevance of myogenin in this system, we have repeated the experiments in the presence of siRNA targeting myogenin. Our results show that MyoD can induce AKAP6 expression via myogenin (see lines 409-422).

In addition to adding new data to the manuscript, we have included a paragraph in the Discussion regarding the relative contributions of MyoD versus myogenin in the regulation of MTOC assembly at the nuclear envelope (lines 544-552). Furthermore, we adapted the model in Figure 8.

4) Please provide a zero time point in regrowth experiments and check nucleation activity in the experiment in Figure 7G. This is needed to argue that an MTOC is present.

Please note, we did not claim that an MTOC is present in Figure 7G but that nesprin-1α and AKAP6β are sufficient to recruit endogenous PCM-1 in the absence of myogenin. However, based on this question, we have now performed regrowth assays in myogenin-negative C2C12 cells after nesprin-1α and AKAP6β co-expression. Our results indicate that there is no significant NE-MTOC activity induced in this setting, whereas MTOC activity was observed at centrosomes. This is in agreement with the fact that myogenin appears to induce centrosome attenuation independent of the presence of nesprin-1α and AKAP6β at the nuclear envelope (see Point 1). The new data are shown in Figure 7 – supplement 1 and described in the text starting at line 497. For regrowth experiments, we have added “zero time points” in Figure 2 – supplemental figure 3.

5) The introduction and discussion should be improved by discussing:– The relevance of the presented findings for muscle function and disease.

As requested, we have added information in regards to the relevance of the NE-MTOC and our data in muscle function and disease to the Introduction (lines 68-77) and Discussion (lines 580-596).

– The roles of the particular centrosomal proteins that are recruited to the nuclear envelope in generating and organizing microtubules.

Information regarding the roles of the different MTOC proteins has been added to the Introduction (lines 104-117).

– The term "centrosomal proteins". As they shuttle from centrosome to the nuclear envelope, the term "MTOC proteins" or similar may be more appropriate.

As suggested, the term “centrosomal proteins” has been replaced with “MTOC proteins” throughout the manuscript.

Reviewer #3:The manuscript by Becker et al. is a new submission of previously eLife submission that focusses on the transition from centrosomal MTOCs to non-centrosomal MTOCs, specifically the surface of the nucleus. This submission more clearly focuses on triggering ncMTOC transition in developing skeletal muscles via the function of MyoD and Myogenin, members of the transcription factors family termed myogenic regulatory factors (MRFs). The final model shown in Figure 8 is well supported by the presented data, which I believe is much improved over the previous manuscript. Reading the recently published work from the same group (Vergarajauregui 2020), which appears to have been in review at the same time as the original submission of this manuscript, helped clarify some of the interaction and functional claims downstream of AKAP6. The authors have done a great job addressing all my previous concerns, they should be commended for this and my comments below are mostly small changes that do not move the goal line for the authors.1) In several cases in the beginning of the manuscript (ex. First paragraph pf the results) it was not clear that ncMTOC specifically referred to nuclear envelope MTOCs. I would recommend explicitly stating in this first result section that throughout the manuscript, ncMTOCs refers to the nuclear envelope in this study. Alternatively, maybe the authors can use NE-MTOC to distinguish it from other ncMTOCs.

Thank you for this recommendation. We have replaced as suggested “ncMTOC” with “NE-MTOC” in the manuscript where appropriate.

2) On line ~181 the authors state that "both MRFs are sufficient individually to induce a switch from centrosomal MTOC to a nuclear envelope MTOC." At this point in the manuscript the authors had not yet shown the elimination of centrosomes, that comes in figure 3. I suggest removing a reference to a switch at this point.

The text has been changed accordingly.

3) In several location in the manuscript the author state that myogenin is required to recruit centrosomal proteins. This can be found in lines 193 and 570 and other places. It is more accurate to say that myogenin is require for the localization of centrosome proteins to the nuclear envelope. Recruitment implies myogenin is the recruiting protein, which it is not, it is simply upstream in the ncMTOC formation pathway.

Thank you for pointing out this issue. The text has been changed accordingly.

4) The figures overall are much improved and the quantification is suitable. However, It is important to report the Ns in each of the experiments. For example in figure 3, how many nuclei were measured? How many experimental repeats were performed.

The requested information has been added.

5) Presentation of the analysis leading up to AKAP6 was great

Thank you.

6) How reliable is the specificity of the E-Box PCR amplification? Have the authors confirmed specificity of the ChIP experiment by PCR amplifying other regions of the AKAP and nesprin promotors? Although, it does appear to be specific enough to distinguish the different isoforms.

To assess the specificity of the ChIP experiments, we have performed additional experiments. For details, see response to essential revisions, point 3.

7) Direct protein interaction cannot be determined by IPs. The authors should change the conclusion that "AKAP6 localizes to the nuclear envelope in differentiated skeletal muscle cells via interaction with nesprin-1α" to AKAP6 localization depends on nesprin-1a. An interaction can only be determine using purified proteins or Y2H as the authors did with AKAP6 and PACT in the (Vergarajauregui 2020). If there is direct protein interaction data, the authors should present it, otherwise the current results should be presented in terms of being upstream or downstream.

We agree with the reviewer and have changed the text accordingly.

Another example of this is "AKAP6 is required for the localization of centrosomal proteins to the nuclear envelope, in part by interacting with PCM-1." If there is no direct protein interaction data available (here or in the literature), then the word interacting should be replaced by recruiting.

The text has been modified accordingly.

[Editors’ note: further revisions were suggested prior to acceptance, as described below.]

Suggested revisions:The paper is very much improved and you have addressed all the remaining points raised by the reviewers. I only have a couple of suggestions before formal acceptance.1) It seems odd to not mention your recent work on AKAP6 published in eLife in the introduction and the result, where you describe identification of AKAP6. I would include statements referring to this work.

We included information about our recent *eLife* paper on AKAP6 in the Introduction and Results (lines 117-122 and 342-344).

2) You showed that recombinant nesprin and AKAP6 expression is sufficient to recruit MTOC protein to the NE, but your new data now show that this is not sufficient to generate an active MTOC. In my opinion this finding is very important and deserves a discussion. It suggests that myogenin induces another factor that is crucial for MTOC activity. While you correctly state "recruitment of MTOC proteins" rather than "MTOC formation" in the paper, non-expert readers may assume that this is synonymous. Without clarification/discussion this could be misleading.

We added a paragraph to the Discussion to clarify the results of the AKAP6β/nesprin‑1α co-expression experiment (lines 586-597).